# Reconstructing the three-dimensional architecture of extrachromosomal DNA with ec3D

Biswanath Chowdhury [1,15,18], Kaiyuan Zhu [1,18], Chaohui Li [1,18], Jessica Alsing [2], Jens Luebeck [1], Maria E. Stefanova[3], Owen S. Chapman[4,5,16], Katerina Kraft[6,7,17], Shu Zhang[6,8,9], Jun Yi Stanley Lim[10], Yipeng Xie[10], Yoon Jung Kim[10], Sihan Wu [10], Lukas Chavez[4,5], Guy Nir [2], Anton G. Henssen [3,10,11,12], Paul S. Mischel [9,13], Howard Y. Chang [6,7,17] & Vineet Bafna [1,14] ✉

Extrachromosomal DNAs (ecDNAs) are circular DNA molecules prevalent in human cancers that drive tumor evolution and drug resistance. Their circular topology, which disrupts topological domains and rewires regulatory circuits, has typically been studied via pairwise interactions. Here we develop ec3D, a computational method for reconstructing three-dimensional ecDNA structures from Hi-C data. Given a candidate ecDNA sequence and whole-genome Hi-C data, ec3D reconstructs spatial structures by maximizing the Poisson likelihood of observed interactions. We validate ec3D using simulated structures, previously characterized cancer cell lines, and microscopy imaging. Our reconstructions reveal that ecDNAs occupy spherical configurations and mediate unique long-range regulatory interactions involved in gene regulation. Through algorithmic innovations, ec3D can resolve complex structures with duplicated segments, identify multi-way interactions, and identify potential intermolecular (trans) interactions. Our findings provide insights into how ecDNA's spatial organization bypasses normal chromosomal constraints and contributes to increased oncogene expression.

Somatic copy number amplification of oncogenes is a major driver of cancer pathogenicity[1–3]. Recent studies[4–6] have revealed that oncogenes are often amplified by extrachromosomal DNA (ecDNA). EcDNAs are highly prevalent, occurring in approximately 15% of early-stage and 30% of late-stage cancers[7], but are rarely seen in normal cells[6]. The presence of ecDNA in tumors is associated with increased pathogenicity and poor outcomes for patients[6]. While this can partially be attributed to increased oncogene expression associated with copy number amplification on ecDNA, recent results point to other contributing factors. EcDNAs have highly accessible chromatin, and their constituent genes are highly expressed, even after accounting for higher copy numbers[5,6].

In normal chromosomes, Topologically Associating Domains (TADs), often bounded by CTCF binding sites, demarcate the regulatory elements that are accessible to a gene[8,9]. In many cancers, the integrity of TADs can be altered[10]. EcDNA formation, which often involves the joining of distal genomic segments, changes chromatin conformation and disrupts existing topological domains, allowing for enhancer hijacking and rewiring of regulatory circuitry[11–13]. EcDNAs often cluster into hubs promoting *trans* regulatory interactions between different ecDNA molecules[14]. EcDNAs with no protein-coding genes have been identified, suggesting an exclusively regulatory role in promoting oncogenesis[15]. Finally, ecDNAs are also suggested to act as roving enhancers for chromosomal genes[16].

Despite the large ($10^5$-$10^8$ bp)[6,17] size of ecDNA, their genomic compositions, including genes and regulatory elements, can be reliably identified using short and long read whole genome sequencing[18–21]. However, a deeper understanding of the regulatory machinery depends not only on the genomic architecture but also on the 3-dimensional conformation of the circular structure. The spatial organization and the three-dimensional structure of ecDNA have, to our knowledge, not been investigated previously.

High-throughput chromosome conformation capture technique (Hi-C) is a dominant technology for characterizing the 3D genome organization[22–24], identifying TADs, and understanding long-range chromatin interactions[8]. The technique quantifies the interaction frequency between each pair of genomic loci, presented in the form of a 2-dimensional matrix. High frequency correlates with spatial proximity, which can be attributed to (a) genomic proximity, (b) structural variation that brings distal loci together, and (c) topological constrictions in DNA structure. Computational methods have been developed to identify significant pairwise interactions suggesting spatial proximity of pairs that are distant in the reference chromosomes[25–27]. While they provide important structural and topological information, Hi-C is a 2-dimensional projection of the 3-dimensional structure, and some important structural features are not immediately discernible. Therefore, these methods do not typically identify multi-way interactions or interactions induced by structural variation, with few exceptions (such as NeoLoopFinder[28]). Smaller changes in 3-dimensional configuration are not immediately apparent in the Hi-C projection. Finally, none of the existing methods accounts for the circular topology of ecDNA.

Many recent methods have been developed to infer the 3-dimensional structure directly from Hi-C data with increasing resolution, and they have been applied to large genomic segments including human chromosomes[29–36]. However, cancer genomes and ecDNAs in particular present unique challenges for these methods. Most ecDNAs involve complex structural variations, joining together genomic segments from different chromosomes. They may also contain multiple copies of large genomic segments, showing aggregated signals of interactions in the Hi-C matrix, which must be implicitly or explicitly de-duplicated.

In this work, we present ec3D, which reconstructs the three-dimensional structure of ecDNA using deep Hi-C data and identifies topological constrictions and clusters of statistically significant chromatin interactions, including multi-way and crossing (non-planar) interactions. We use ec3D to reconstruct the 3-dimensional structures of ecDNAs in multiple cancer cell lines and use the structures to better characterize the unique regulatory biology of ecDNA. The 3D structures allow for improved detection of CTCF binding, A/B compartmentalization, and enhancer-promoter interactions. They also point to putative sites of ecDNA-protein, ecDNA-ecDNA interactions, and ecDNA-chromosome tethering, thereby providing valuable avenues for exploring ecDNA biology.

## Results

### Overview of ec3D

Ec3D uses two types of data: (i) a local assembly of ecDNA sequence and (ii) a whole-genome Hi-C contact matrix, both aligned to the same reference genome (see **Methods** for how these data can be obtained). The input ecDNA sequence is represented by ordered and oriented genomic segments in Browser Extensible Data (BED) format, possibly with segments occurring multiple times. The Hi-C matrix describes the interaction frequencies for pairs of bins, each representing a genomic region of pre-specified resolution (default 5 kb), in either *hic* or *cool* format.

With these inputs, ec3D first extracts Hi-C submatrices corresponding to segment pairs, where both segments are chosen from the ecDNA sequence. Ec3D reassembles these submatrices into a single matrix $C$ of dimension $N_c \times N_c$ bin pairs, representing chromatin interactions within ecDNA intervals (Fig. 1). Next, ec3D reconstructs the 3D structure of the input ecDNA by maximizing the joint Poisson likelihood[30,32], which models interaction frequencies $C_{ij}$ as independent Poisson random variables with mean $\lambda = \beta d_{ij}^\alpha$, a decreasing function of the Euclidean distance $d_{ij}$ between bin $i$ and bin $j$, with a scaling

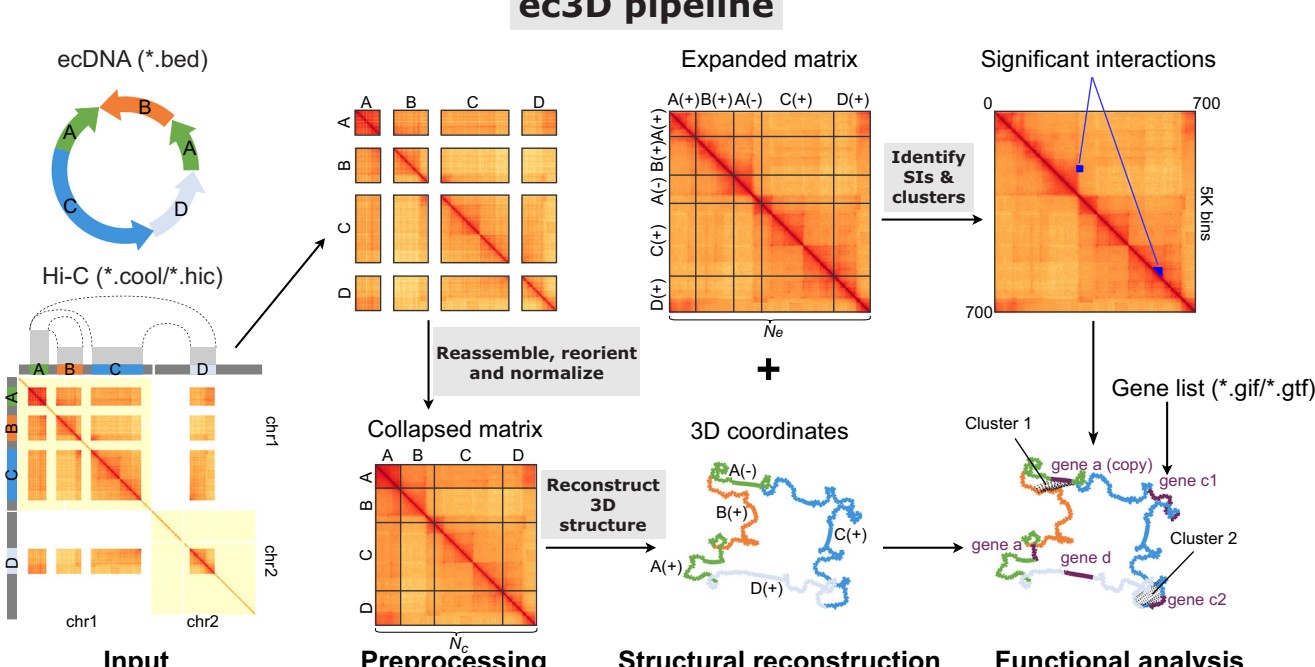

**ec3D pipeline**

**Fig. 1 | Overview of the ec3D workflow.** Ec3D takes the amplicon sequence coordinates and chromatin capture (Hi-C) data as input, and outputs the 3-dimensional coordinates of each fixed-resolution bin. It also resolves the structure of duplicated regions within ecDNA. Finally, it computes and reports significant interactions between pairs of bins.

parameter $\beta > 0$ and a power law decay parameter $\alpha < 0$ (see **Methods** for details).

A key feature of ec3D is reconstructing ecDNAs containing duplicated segments. In such cases, some entries in the Hi-C matrix have interaction data from multiple bins (duplicated intervals) in the reconstruction, and this matrix is named a *collapsed* matrix. Ec3D automatically reconstructs a structure with $N_e$ 3-dimensional coordinates, one for each (potentially duplicated) fixed-resolution bin. If duplicated bins exist, $N_e$ exceeds the dimension $N_c$ of the input Hi-C matrix. The observed interaction frequencies are modeled as the sum of Poisson random variables from all copies, enabling duplicated segments positioned in 3D space. Otherwise, $N_e = N_c$ and the Poisson model directly applies.

After determining the 3D structure of ecDNA, ec3D constructs an *expanded* Hi-C matrix $E$ of dimensions $N_e \times N_e$ by redistributing the interactions to individual copies of bin pairs, proportional to their spatial distance in the reconstructed structure (see **Methods**). Next, Ec3D identifies significant interactions within this expanded matrix (Fig. 1). These significant interactions are subsequently clustered using the Louvain method[37]. Ec3D outputs the expanded Hi-C matrix corresponding to the ecDNA sequence, the reconstructed 3D structure coordinates as a text file (Fig. 1), and an interactive structure visualization showing associated genes and clusters of significant interactions.

### Ec3D reconstructs structures accurately on simulated data

Given a ground truth structure and the corresponding expanded or collapsed Hi-C matrix, ec3D can reconstruct a 3D structure with the Hi-C matrix, and its performance can be measured by comparing the ground truth and reconstructed structures. We developed an extensive suite of simulated ecDNA structures and Hi-C matrices to benchmark ec3D's performance. Very briefly, we simulated *base* structures with $k$ ($k \in \{1, 2, 3\}$) *topological constrictions (TCs)*. (See Methods and Supplementary Methods for details.) Each TC corresponds to a pair of genomic regions that are genomically distant but spatially close. We also added multiple random local folds to the base structures (Supplementary Fig. 1). Structures, which share the same topological constrictions but differ in local folds, are referred to as having the same base structure. Each simulated structure is described as an $N_e \times 3$ matrix (Fig. 2a), corresponding to the 3D coordinates of $N_e$ bins. We simulated 30 random structures for each value of $k$ (the number of TCs), resulting in a total of 90 simulated 3D structures.

For each simulated structure, we generated 10 simulated Hi-C matrices (Fig. 2b) by sampling interaction frequencies from the Poisson distribution described above, with random combinations of $\alpha \in [-3, -0.75]$ and $\beta \in [1, 10]$, which cover a typical range we observed in real data. This gives 900 simulated Hi-C in total. The first 450 $N_e \times N_e$ matrices $E$ are expanded matrices with duplicated bins kept separate. The other 450 $N_c \times N_c$ matrices $C$ are collapsed matrices after merging duplicated bins. To simulate Hi-C with duplicated bins, we randomly selected two regions, each with $l$ bins, as duplicated regions. We then generated the collapsed Hi-C matrix $C$ by summing the interactions for the duplicated regions from the original expanded matrix $E$. Thus, if the original sample with $N_e$ bins had $l$ bins duplicated, the dimensionality of the collapsed matrix $C$ became $N_c \times N_c$, where $N_c = (N_e - l)$. Note that the 3D structures were not changed when collapsed matrices were generated from expanded matrices. In a simulated 3D structure, topological constrictions and local folds contribute to global and proximal interactions in $E$, which mimic the Hi-C matrix of a real ecDNA sample.

To evaluate performance, we measured the root mean square deviation (RMSD) and the Pearson correlation coefficient (PCC) between the ground truth and the structures reconstructed from the Hi-C matrix (Fig. 2a–c). The RMSD compares coordinates of two structures and requires a rotation and translation step for optimal

alignment, whereas PCC is measured directly on pairwise distances between pairs of bins (see **Methods**). The median RMSD values of the 450 reconstructions without duplication was 0.058, with an interquartile range IQR = [0.032, 0.106], which was significantly lower than RMSD values computed both by comparing two randomly selected structures with the same base structure (median RMSD 0.338, IQR = [0.268, 0.429], Wilcoxon rank-sum test, p-value = 3.8e-122), and by comparing two random structures with different base structures (median RMSD 0.573, IQR = [0.525,0.638], Wilcoxon rank-sum test, p-value = 1.2e-147) (Fig. 2d; Supplementary Data 1, 2). This result suggested that ec3D can reconstruct 3D structures with high accuracy and even reconstruct smaller local folds accurately. Similar results were seen with the PCC metric - the PCC values for the reconstruction were significantly higher than those computed by comparing random structures (Supplementary Fig. 2a; Supplementary Data 1, 2). Notably, samples with $k = 2$ and 3 topological constrictions had lower median RMSD values and higher median PCC, compared to samples with $k = 1$ (Supplementary Fig. 3). This improved performance was likely due to stronger global interactions in samples with a higher number of constrictions, resulting in more constraints on possible structures.

We next evaluated the ability of ec3D to reconstruct structures with duplicated bins. We ran ec3D on the 450 collapsed matrices and obtained the median RMSD 0.102 (IQR = [0.054,0.201]), which again was significantly better than two random structures with the same base structure (Wilcoxon rank-sum test, p-value = 5.5e-102) and with different base structures (Wilcoxon rank-sum test, p-value = 2.0e-148) (Fig. 2d; Supplementary Data 1, 2). Comparisons using the PCC metric were highly correlated with RMSD (Supplementary Fig. 2a). Note that it is not known in advance if the duplicated regions fold into a similar local substructure. Therefore, in our simulations, we selected half of the samples to have the same local substructure in the duplicated regions, while the other half had different local substructures. The RMSD and PCC values in the two cases were very similar (Supplementary Fig. 4), indicating that ec3D has consistent performance regardless of the similarity of local substructures in the duplicated regions.

Because the raw RMSD and PCC values are data dependent and difficult to interpret directly, we compared the PCC (respectively, RMSD) value of ground truth versus a reconstructed structure against the PCC (RMSD) values of the ground truth versus a random structure. The vast majority (97.56%) of reconstructed structures had higher PCC than random structures (Fig. 2e). Similarly, 95.72% of reconstructed structures had lower RMSD than random structures (Supplementary Fig. 5).

Next, we tested the accuracy of ec3D estimates of the power law decay parameter $\alpha$ by measuring the correlation between the true and estimated values of $\alpha$ in the 900 reconstructions. The ground truth and estimated values were highly correlated (Supplementary Fig. 6). Defining the error as $\frac{1}{n}\sum_{i=1}^{n}|\hat{\alpha}_i - \alpha_i|/\alpha_i$, where $\alpha_i$ is the ground-truth, and $\hat{\alpha}_i$ the estimated value for sample $i$, the mean error values in estimating $\alpha$, for samples without and with duplication, were 2.32% and 3.68%, respectively. The results indicated that $\alpha$ could be estimated accurately in most samples, regardless of duplication. The estimation accuracy was higher when the true $\alpha$ values were large ($\simeq$-1). To investigate this further, we reanalyzed the RMSD values of all 900 simulated samples across the different ranges of $\alpha$ values. We found that structure reconstruction accuracy was also better on samples with larger $\alpha$ values (Fig. 2f). Notably, $\alpha$ values of real data obtained from human samples tend to be close to -1 (Supplementary Data 3), further raising confidence in the accuracy of our reconstructions on real data.

Expectedly, the objective value (negative log-likelihood) decreased smoothly with iterative optimization until convergence. Broadly, the RMSD (respectively, PCC) metric also decreased (respectively, increased), but the transition was much sharper so that a

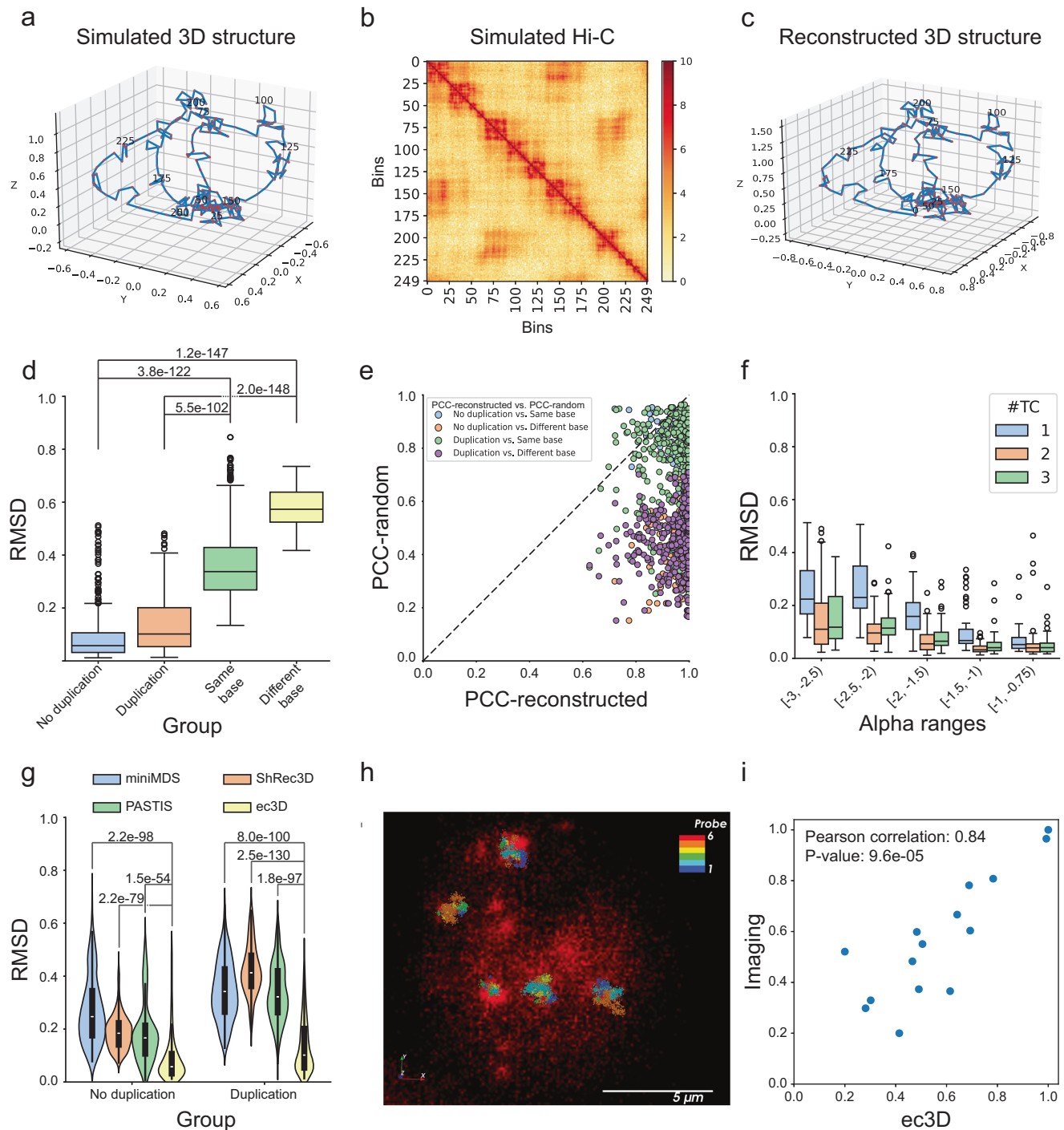

**Fig. 2 | Performance of ec3D on simulated data. a** A simulated 3D circular structure (ground truth) with 250 bins. **b** A simulated Hi-C matrix generated from the structure in (**a**). **c** The reconstructed structure computed by running ec3D on the Hi-C data in (**b**). **d** RMSD-metric simulation test results in 4 different groups: No duplication - ground-truth versus reconstructed structures without duplication; Duplication - ground-truth versus reconstructed structures with duplication; Same base - random pairs of structures with the same base structure; Different base - random pairs of structures with different base structures. Each group has 450 pairs of samples. P-values were calculated using a one-sided Wilcoxon rank-sum test for two samples. Center lines indicate the median. Boxes represent the interquartile range (IQR) from the 25th to the 75th percentile. Whiskers extend to the minimum and maximum values within 1.5 times the IQR (the same below). **e** PCC values of ground truth versus reconstructed structures (PCC-reconstructed) compared to PCC values of ground truth versus random structures (PCC-random). A data point at the bottom right of the dashed line indicates that the reconstructed structure is

more similar to the ground truth than a random structure. **f** Distribution of RMSD values (ground truth versus reconstructed structures) over $\alpha$ values. Each range of $\alpha$ value and number of topological constrictions includes 60 samples (30 with duplication and 30 without duplication). **g** Violin plots showing the RMSD comparisons between ec3D and other methods that reconstruct DNA structure from Hi-C. Box plots within each violin indicate the median, interquartile range (IQR) from the 25th to the 75th percentile, and whiskers extending to the minimum and maximum values within 1.5 times the IQR. P-values were calculated using a one-sided Wilcoxon rank-sum test for two samples. **h** Imaged TR14 ecDNAs using Sequential OligoSTORM. **i** The correlation of the pairwise bin distances obtained by ec3D and by OligoSTORM imaging averaged across 10 ecDNA molecules. The distances were normalized to a range of 0.2 –1.0 by min-max normalization. The P-value was calculated using a two-sided t-test. Source data are provided as a Source Data file.

relatively modest improvement in the beginning was followed by a more dramatic shift later (Supplementary Fig. 7). Intriguingly, the initial and final RMSD and PCC values of all runs (900 samples × 5 repeats) were positively correlated, with correlation scores 0.704 and 0.571, respectively (Supplementary Fig. 8), highlighting the importance of the initialization step in reaching optimal final structures.

In our model, ecDNA chromatin forms a continuous polymer chain, where fixed resolution bins are represented as discrete points along the chain. To ensure two adjacent bins have roughly equal spatial distance in the 3D space, we included a regularization term (see **Methods** and Supplementary Fig. 9). The impact of this additional constraint on reconstruction was modest. The average absolute difference between the RMSD (and PCC) values with and without regularization was 0.0324 (and 0.0170) in the simulated data.

## Ec3D compute time

All samples were run on a supercomputing node equipped with two 64-core AMD EPYC 7742 processors and 256 GB of DDR4 memory, with at most 16 threads and 2 GB of memory allocated for each sample. Because ec3D follows a stochastic optimization function, its running time varies from sample to sample. Running time increased with the number of bins (Supplementary Fig. 10a). Duplications took a longer time to resolve (Supplementary Fig. 10b). Most ($\geq 90\%$) samples without duplication could be resolved within 12,000 s, and most samples with duplication could be resolved within 35,000 s.

## Ec3D versus other reconstruction methods

We compared ec3D against three existing 3D genome reconstruction methods, MiniMDS[35], ShRec3D[31], and PASTIS[32], with the caveat that the other methods were not specifically developed for ecDNA and cannot handle structural variations and duplications. MiniMDS and ShRec3D utilize multidimensional scaling with fixed parameters, while PASTIS models a joint Poisson likelihood to simultaneously infer structure and parameters and can be considered as the starting point of our work. Using both metrics, ec3D significantly outperforms the other methods, and the performance gap is wider in samples with duplicated bins (Fig. 2g, Supplementary Fig. 2b).

## Ec3D predicted structures correlate strongly with imaged ecDNAs

High-resolution microscopy using fluorescent probes has also been used to understand the 3-dimensional structure of DNA molecules[38–40], but has not been attempted previously for an entire ecDNA. Notably, the process is not trivial because each cell contains an unknown number of ecDNA molecules. To provide a baseline comparison, we selected the TR14 cell line, which amplifies the oncogene *MDM2* on ecDNA. We acquired Hi-C data and reconstructed the 3D structure using ec3D (see next section). Next, we designed custom probes covering the ecDNA region with 200-kb genomic resolution. The probes were imaged using Sequential OligoSTORM[38,41] (see **Methods**). The centroids of fluorescent signals from each probe were used to identify putative ecDNAs in a cell (Fig. 2h **and** Supplementary Fig. 11a–c). Pairwise distances predicted by the OligoSTORM images strongly correlated with the corresponding pairwise distances predicted by ec3D (Pearson Correlation coefficient 0.84, Fig. 2i), providing a baseline, orthogonal validation of ec3D structures.

## Ec3D reveals circular structure of ecDNA linking distant segments

We applied ec3D to high coverage Hi-C data acquired from 9 cancer-derived cell lines (Supplementary Data 3). 7 of the 9 cell lines contained ecDNA, while the other two, GBM39HSR and IMR-5/75, contained intrachromosomal focal amplifications that displayed as Homogeneously Staining Regions (HSRs). We used previously published reconstructions of the ecDNA and HSR sequences to obtain the genomic regions of the amplicons (**Methods;** Supplementary Data 4).

Scatter plots comparing Hi-C interaction frequency and 3D distance showed a clear inverse relationship on a log-log scale, confirming the expected negative power law decay relationship between frequency and distance in 3D space suggested by the Poisson model (Fig. 3a, Supplementary Fig. 12). The correlation was very strong, with PCC ranging from -0.97 to -0.76. Notably, the correlation magnitude increased with increasing Hi-C contact (Supplementary Fig. 12). The results indicate a more consistent and precise prediction of spatial distances as interaction frequencies increase. The observed horizontal scatter of bins for low distances was due to the regularizer term, which forced adjacent bins to have similar Euclidean distances even if their interaction frequencies varied.

Previous estimates[22,32] of $\alpha$ range from $\alpha \simeq$ -3 to $\alpha \simeq$ -1.5. The optimal values of $\alpha$ on ecDNA structures were somewhat larger, estimated as -1.05 ± 0.27 (Supplementary Data 3). The significantly smaller decay of interaction strength with increasing Euclidean distance suggests that ecDNAs maintain their structures despite their large size and volume.

All ecDNA reconstructions naturally converged to circular 3-dimensional structures in contrast with the structure of identical regions in control cell lines. For example, for the GBM39 ecDNA, a relatively simple structure was formed by a single front-to-back joining of a chr7 segment that encompasses the oncogene *EGFR* (Fig. 3b). High spatial proximity between the first and last bin was automatically discovered by ec3D. For comparison, we reconstructed the structure of the identical genomic region in GM12878, a cell line where *EGFR* is located on the chromosome (Fig. 3c). The reconstruction on GM12878 showed similarity in the smaller topological domains, but importantly, no interactions between the first and last bins.

## EcDNA structures are oblate spheroidal and occupy all three dimensions

Scanning electron microscopy data on cultured cells in metaphase[5] do not reliably explain whether ecDNAs occupy a sphere-like or a disk-like volume. To address this question, we first computed a minimum volume bounding cuboid that captured the overall shape of the reconstructed 3D structure (Fig. 3d). Had the 3D structure of ecDNA been disk-like, we would expect the smallest dimension of the cuboid to be much smaller than the largest dimension. However, the ratios between the minimum and maximum edge lengths of the bounding box of the ecDNA structures were generally high, ranging from 0.476 (GBM39) to 0.895 (H2170) (Supplementary Data 3). This suggested that ecDNA structures were oblate spheroidal with a large third dimension.

We next tested if the ecDNA could be embedded in a flatter bounding box (i.e., with smaller edge length ratios) and still generate the observed Hi-C interactions. Specifically, we reconstructed 3D structures of the GBM39 ecDNA (amplifying *EGFR*) and RCMB56 ecDNA (amplifying *DNTTIP2*) by fixing the parameter $\beta$ with optimal estimated values ($\beta$=4 for RCMB56 and $\beta$=16 for GBM39) and repeatedly halving the maximum range in the first axis without modifying the range [-1, 1] of the other two axes. By fixing the scaling factor $\beta$, we ensured that the structure was not shrinking proportionally in all axes in reconstruction. We hypothesized that for disk-like structures, decreasing the range of one axis would not impact the Poisson likelihood, as bins could still be placed on a plane orthogonal to that axis, preserving the pairwise spatial distances; however, for spherical structures, the Poisson likelihood would become worse, due to additional constraints in the 3D space disrupting expected spatial distances suggested by Hi-C interactions. For GBM39, the likelihood indeed became worse as the smallest dimension decreased from 0.25 to 0.12 (Fig. 3e, Wilcoxon rank-sum

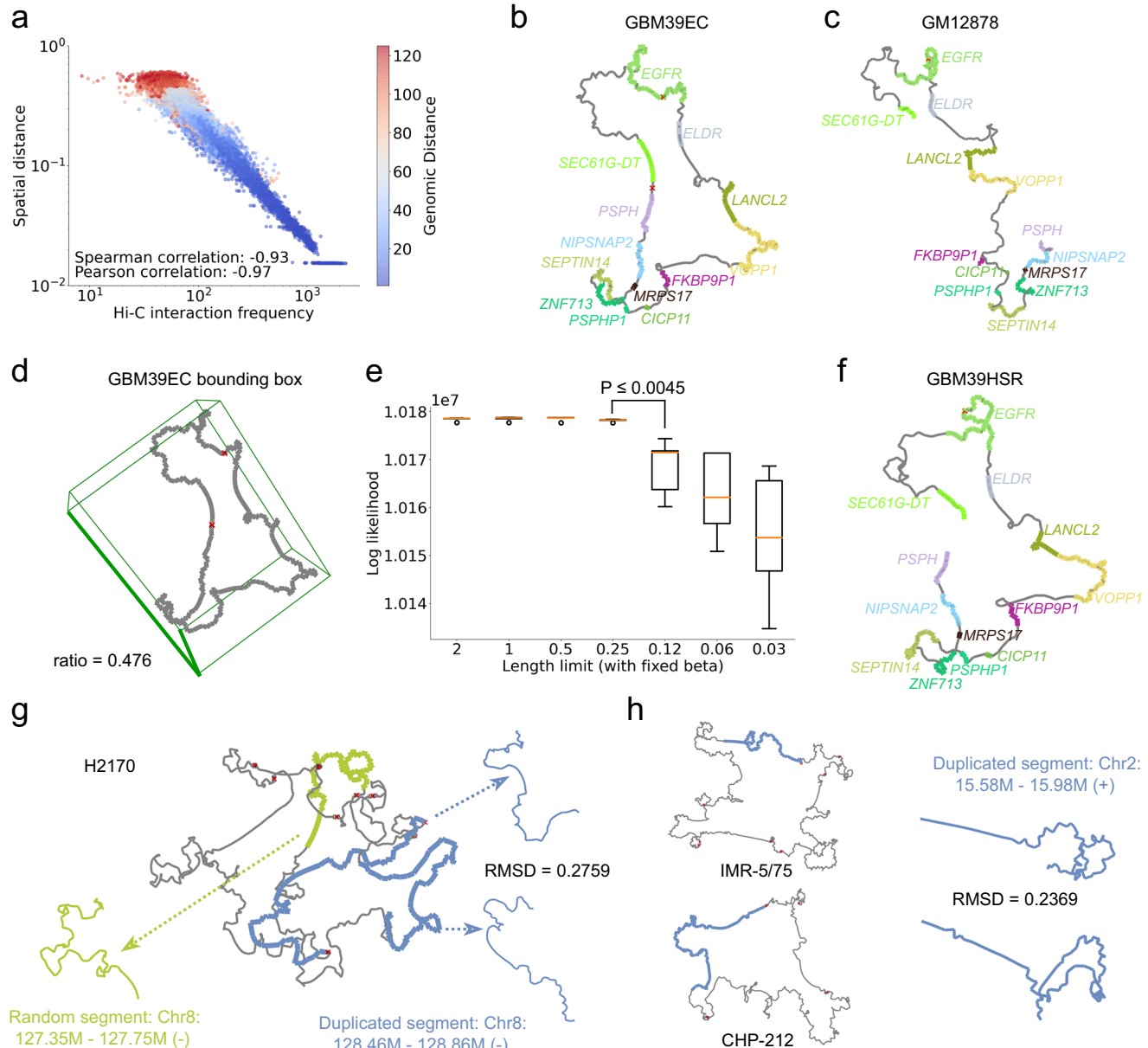

**Fig. 3 | Structural properties of ecDNA. a** Correlation between normalized Hi-C interaction frequencies (x-axis) and spatial (Euclidean) distances from ec3D reconstruction (y-axis) from GBM39. Each point in the plot corresponds to a pair of 5-kb bins. Color gradient representing the genomic distance was overlaid on each scatter plot point, with warmer colors indicating shorter genomic distances and cooler colors indicating longer distances. The Spearman and Pearson correlation coefficients suggested a negative power law decay of interaction frequencies with spatial distances. **b** 3D structure of GBM39 ecDNA with oncogenes amplified on the ecDNA. Genes are highlighted, and red crosses represent structural variations leading to ecDNA formation. **c** 3D structure of the same chromosomal segment (chr7:54.7–56.1 Mb) on a control cell line, GM12878, reconstructed by ec3D. **d** Size of the minimum volume bounding box enclosing GBM39 ecDNA, suggesting an oblate spheroidal structure. **e** Optimal values of ec3D's objective function (y-axis) after

fixing the scaling parameter $\beta$ and limiting the maximum range of the first axis to force a flatter structure (x-axis). A box plot describing the 5 final objective values with random initialization $X$ was made for each length limit ranging from 0.03 to 2. P-value was calculated using a one-sided Wilcoxon rank-sum test for two samples. Center lines (orange) indicate the median. Boxes (black) represent the interquartile range (IQR) from the 25th to the 75th percentile. Whiskers extend to the minimum and maximum values within 1.5 times the IQR. **f** 3D structure of GBM39HSR reconstructed by ec3D. **g** 3D structure of H2170 ecDNA. The duplicated segment chr8:128.4–128.9 Mb showed significant similarity (RMSD = 0.276, permutation test p-value = 0.01, see **Methods**). **h** A segment from chromosome 2:15.58M–15.98 M amplified on two different cyclic (ecDNA/HSR) structures with significant similarity (RMSD = 0.237, permutation test p-value = 0.007, see **Methods**). Source data is provided as a Source Data file.

test, p-value ≤ 0.0045). For RCMB56, the likelihood reduced significantly as the smallest dimension decreased from 1 to 0.5 (Supplementary Fig. 13, p-value ≤ 0.0045), strongly suggesting a spheroidal conformation. Our results are consistent with ecDNA requiring all 3 dimensions for optimal folding, providing additional freedom for complex topological constrictions.

## Ec3D reveals high structural similarities between HSR and ecDNA in isogenic lines

The cell line GBM39HSR is isogenic to GBM39EC but with an intrachromosomal or HSR amplification of EGFR. Remarkably, the Hi-C pairwise interactions of the amplified region were highly similar (Correlation = 0.9859, Supplementary Fig. 14a–c). Previous findings

have suggested that HSRs can be formed via reintegration of tandemly duplicated copies of ecDNA into a chromosomal locus[42], and this is supported by the similarity of the breakpoints in the isogenic cell lines.

To rebuild the structure of GBM39HSR, we duplicated the first 3 bins (**Methods**) during preparation of the collapsed matrix and ran ec3D using this genome with duplications. The 3D reconstructions of GBM39EC and GBM39HSR were also remarkably similar (Fig. 3b, f, Supplementary Fig. 14d), with RMSD 0.346 (PCC = 0.937). By comparison, the RMSD between identical regions in GM12878 and GBM39EC was higher at 0.394 (PCC = 0.850). The similarity between GBM39EC and GBM39HSR structures matched that of *random* structures with the same base structures (median RMSD = 0.338; Fig. 2d), suggesting that the major topological constrictions were identical, but ec3D captured fine structural differences between the ecDNA and HSR structures in a way that the Hi-C image could not (Supplementary Fig. 14d–f). For example, the spatial distance between chr7:54.865-54.87 Mb and chr7:55.08-55.1 Mb nearly doubled from 0.16 in GBM39EC to 0.3 in GBM39HSR.

Despite these advances, the current Hi-C data do not provide enough resolution to distinguish between different possible HSR substructures. Distinct structures, such as the 'spring' and the 'petal' shapes (Supplementary Fig. 15), are possible in the tandem duplication model, and resolving the fine HSR structure will likely require new technologies.

A tandem duplication model for HSR had previously been suggested for the *MYCN* amplification in the human neuroblastoma cell line IMR-5/75[12]. In the proposed architecture of this amplicon, a neo-TAD joined two genomically remote segments connecting the *ANTXR1* locus (chr2:68.9-69.2 Mb) and *LRATD1* (chr2:14.5-15.1 Mb) locus, consistent with a tandem joining of the last and first segments. Notably, the structure revealed by Helmsauer et al.[12] to have two TADs was based on a collapsed matrix containing duplicated copies of chr2:14.63–15.1 Mb (Supplementary Fig. 16a). Ec3d automatically generated an expanded matrix that resolved the duplicated region. It found that the two TADs were maintained, and that the duplicated copies of chr2:14.63-15.1 Mb were part of a single TAD, with smaller substructures (Supplementary Fig. 16b). We next asked if these duplicated regions folded into similar substructures.

### Duplicated regions on ecDNA can have similar structures

One key feature of ec3D is the reconstruction of ecDNA structures with duplicated segments. Two ecDNA-positive cell lines, D458 and H2170, and one HSR line, IMR-5/75, contained duplicated segments with sizes ranging from 4 bins (20 kb) to 163 bins (815 kb) (Supplementary Data 5). Each segment was duplicated at most two times on the two ecDNAs, including two inverted duplications in D458. We compared the significance of similarity of the local 3-dimensional structure of the duplicated regions using a permutation test (**Methods**). Of the 8 pairs of duplicated regions (6 of size at least 50 kb), 2 pairs in D458 and 1 pair in H2170 had significantly similar structures (Supplementary Data 5), including for example, duplicated bins [18, 96] and [364, 442] on H2170 (Fig. 3g, permutation test p-value ≤ 0.016). The other 4 duplicate pairs did not have significantly similar structures, including the duplicated pair on the IMR-5/75 HSR.

We next compared identical genomic regions chr2:15.585–15.985 Mb amplified in two different cell lines. The region is amplified on ecDNA in the cell line CHP-212, and on an HSR in IMR-5/75 (Fig. 3h). The RMSD value of 0.237 was highly significant (permutation test p-value ≤ 0.0072), confirming that identical genomic sequences folded into very similar local structures despite the very different context. Together, the results suggest that the underlying DNA sequence only provides partial information for reconstructing the structure. Interactions with other factors and nuclear bodies play a role in determining structure.

### Ec3D clarifies the neo-TAD structures, A/B compartmentalization, and oncogene dysregulation in ecDNA

Recent results on Neuroblastoma cell lines revealed a class of *MYCN* amplicons that lacked key local enhancers of *MYCN*, but hijacked distal fragments containing previously discovered super-enhancers known to mediate Neuroblastoma progression[12]. Hi-C data from the cell line CHP-212 revealed the formation of a neo-TAD that connects *MYCN* to distal super-enhancers. We compared the TAD boundaries identified using scaled ec3D distances (Fig. 4a, **lower triangle**) against those identified using Hi-C (Fig. 4a, **upper triangle**). As TAD structures are often supported by CTCF binding sites[8,24], we measured the proximity of TAD boundaries to CTCF binding. Remarkably, the ec3D predicted TAD boundaries explained 53.85% of the top CTCF peaks (**Methods** and Supplementary Data 7, p-value = 4.5e-10) in comparison to the 26.92% explained using Hi-C (Supplementary Data 8, p-value = 0.0029).

We next compared the A/B compartments of the CHP-212 ecDNA that were generated using either the original Hi-C or the spatial distances from the 3D structure (**Methods**). Compared to Hi-C, ec3D generated a correlation matrix with more intense signals. Further, ec3D identified similar but finer A/B compartment structures on CHP-212 ecDNA (Fig. 4b, Supplementary Fig. 17). We also investigated the A/B compartments of the identical genomic regions on a control cell line, GM12878. In direct contrast with GM12878, three distinct regions spanning chr2:15.58-16.07 Mb, chr2:12.46-12.62 Mb, and chr2:12.62-12.75 Mb were in the same compartment on the CHP-212 ecDNA. We next explored the 3D structure of this compartment as predicted by ec3D (Fig. 4c) and the activity of its constituent genes.

Single-cell RNA-seq data of CHP-212 had previously revealed that out of the 6 genes present on ecDNA, 4 were overexpressed (*LPIN1*, *TRIB2*, *DDX1*, and *MYCN*), but the expression of two genes *GREB1*, *NTSR2* remained at basal level (p-value = 1.0e-11), with median expression at least 5X lower than the minimum median expression of the overexpressed genes (Fig. 4d)[43]. The ec3D structure clarifies this observation by revealing multiple topological domains on the A compartment described above. One domain contained *LPIN1, TRIB2, DDX1*, and the other contained *MYCN* along with a super-enhancer, and this active compartment excluded the two basally expressed genes. Intriguingly, *LPIN1* was split in the ecDNA. The 5' half of *LPIN1* was over-expressed along with *TRIB2* and *DDX1*, while *LPIN1* 3' half was excluded and not expressed (Supplementary Fig. 18a). Moreover, the structural variation that split *LPIN1*, brought the *LPIN1* 5' end in proximity to the 3' end of *MIR3681HG*, resulting in expression of a fused transcript (Supplementary Fig. 18b), and also a readthrough event resulting in expression of intronic DNA downstream of the genomic breakpoint (Supplementary Fig. 18c). Finally, the circularization removed the region immediately upstream of *GREB1*, consistent with its reduced expression. Thus, ecDNA can alter the regulation of genes through a combination of structural variation and 3D conformational change.

We also investigated a TAD on the Medulloblastoma cell line D458, which is a 2.5 Mb molecule amplifying the oncogenes *MYC* (chr8) and *OTX2* (chr14) on an ecDNA. Earlier results had suggested that a DNase-hypersensitive region (DHS1)[44] containing a putative enhancer located 80 kb from the *OTX2* gene on chr14 was essential for proliferation of the cell line[17], and it was speculated that DHS1 might be hijacked by *MYC* to drive proliferation. However, DHS1 was found not to influence *MYC* activity on D458[17]; instead, it enhanced *OTX2* expression in other Medulloblastoma cell lines[44]. Ec3D analysis suggests a neo-TAD that includes DHS1, *OTX2*, and the lncRNA *OTX2-AS1*, but not *MYC*, providing more clarity for the observed experimental data (Supplementary Fig. 19). We also noted that an inversion of the *OTX2* region brought *OTX2-AS1* closer to the enhancer on the ecDNA, in contrast to their positioning on the reference genome.

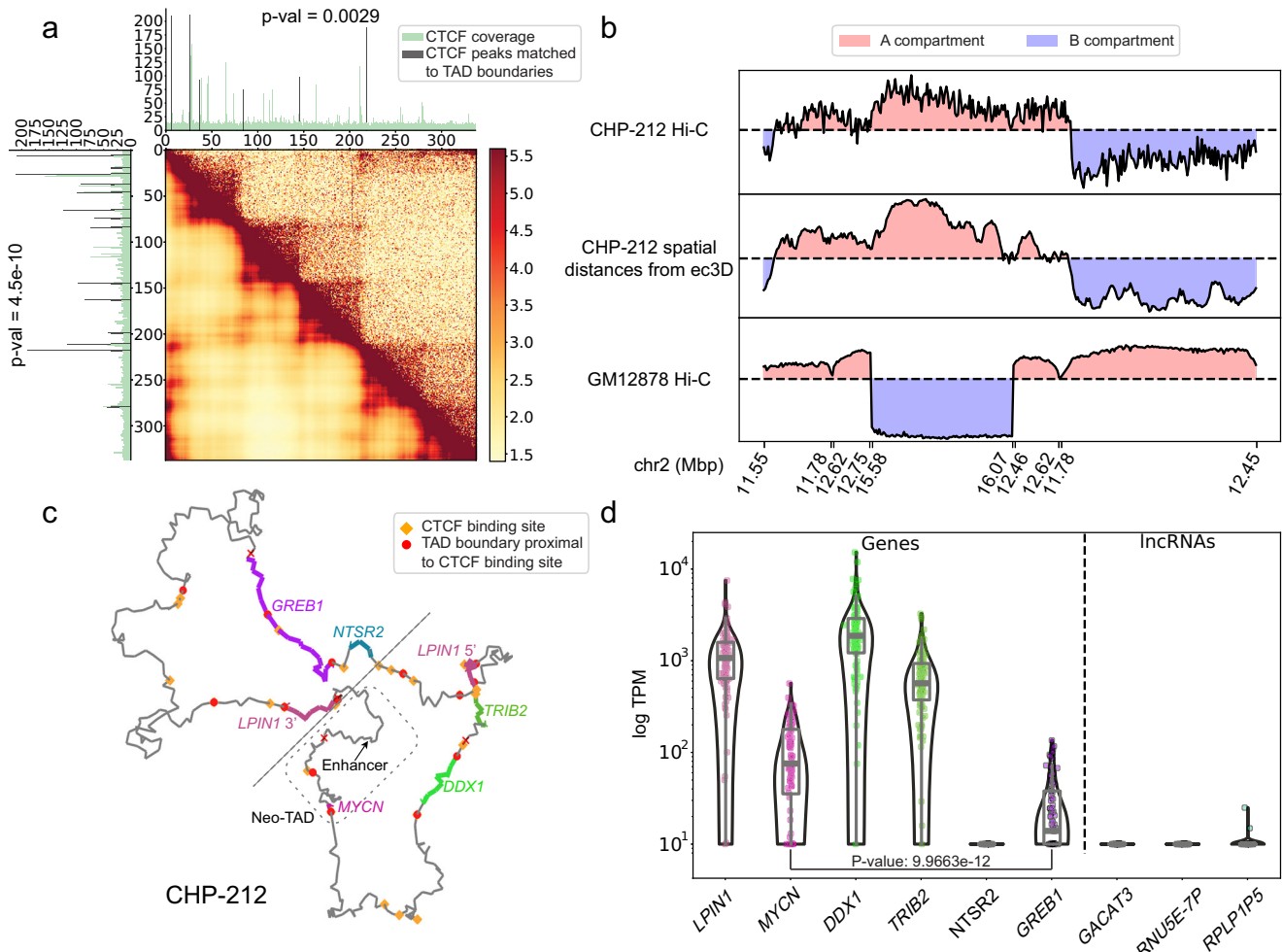

**Fig. 4 | Functional properties of ecDNA clarified by ec3D. a** TAD boundaries identified using scaled ec3D distances (lower triangle) compared against those identified using Hi-C (upper triangle) of cell line CHP-212. The TAD boundaries predicted using scaled ec3D were more proximal to CTCF binding sites (Dark colored peaks). **b** A/B compartmentalization on CHP-212 ecDNA generated using Hi-C and from ec3D reconstructed spatial distances. A/B compartmentalization of the isogeneic region was generated using GM12878 Hi-C as a control. The genomic coordinates were rearranged to match the ecDNA sequence. **c** 3D structure of CHP-212 ecDNA. Dashed box encloses the *MYCN* neo-TAD in CHP-212 containing *MYCN*

and a super-enhancer from a distal segment on chr2. The dashed line distinguishes a region with overexpressed genes (bottom right) from the other region with base-level gene expression (top left). Predicted TAD boundaries proximal to CTCF binding sites are shown. **d** Single cell expression level of genes (left) and lncRNAs (right) amplified on CHP-212 ecDNA, from 96 cells. P-value was calculated using a one-sided Wilcoxon rank-sum test for two samples. Box plots (gray) within each violin indicate the median, interquartile range (IQR) from the 25th to the 75th percentile, and whiskers extending to the minimum and maximum values within 1.5 times the IQR. Source data are provided as a Source Data file.

## The ensemble and consensus ecDNA structures change with biological context

Methods for DNA structure reconstruction generate either a consensus[31,35,45] or an ensemble of structures[34,46,47], with both methodologies being useful for different applications[45]. While ec3D was primarily designed to generate a consensus structure, it can be run with multiple random initializations to obtain an ensemble of structures, which is akin to sampling from a collection of locally optimal structures. To investigate this feature of ec3D, we acquired Hi-C data for the ecDNA-containing MSTO211H cell line in two specific conditions: 2 replicates in G1 phase, and 2 replicates in M phase. For each of these 4 datasets, we generated an ensemble of 5 structures for a total of 20 ec3D-derived structures. The structures revealed some interesting insights. First, the pairwise correlation, measured using PCC, between ensemble structures from any single biological replicate was high (Supplementary Fig. 20). Correspondingly, the pairwise correlation between ensemble structures from two different replicates from the same cell-cycle phase was also high and matched the correlation of ensemble structures within the same replicate in both G1 phase and M

phase. By contrast, the consensus (and ensemble) structures from G1 phase and M phase were very different (Fig. 5a). The results suggest that the consensus structures are a good representation of the ensemble. Additionally, ecDNA structures are governed by external influences such as different ecDNA-protein interactions and ecDNA-chromosome tethering. Ec3D-derived consensus structures can reliably reveal those differences.

## Ec3D reconstructions enable identification and clustering of significant Hi-C interactions

We used ec3D to identify the mechanisms of significant interactions (SIs) between pairs of bins in an expanded matrix. We used 3 methodologies (**Methods**, Supplementary Data 6), each capturing a subset of the possible interactions. Briefly, ref-SI captured SIs relative to expectation on the reference genomes. It was the most general method for capturing significant interactions. The next method, circ-SI, captured SIs after conditioning on the ecDNA sequence, thereby removing interactions due to the joining of distal segments (structural variations) leading to ecDNA formation. The third measure, spatial-SI,

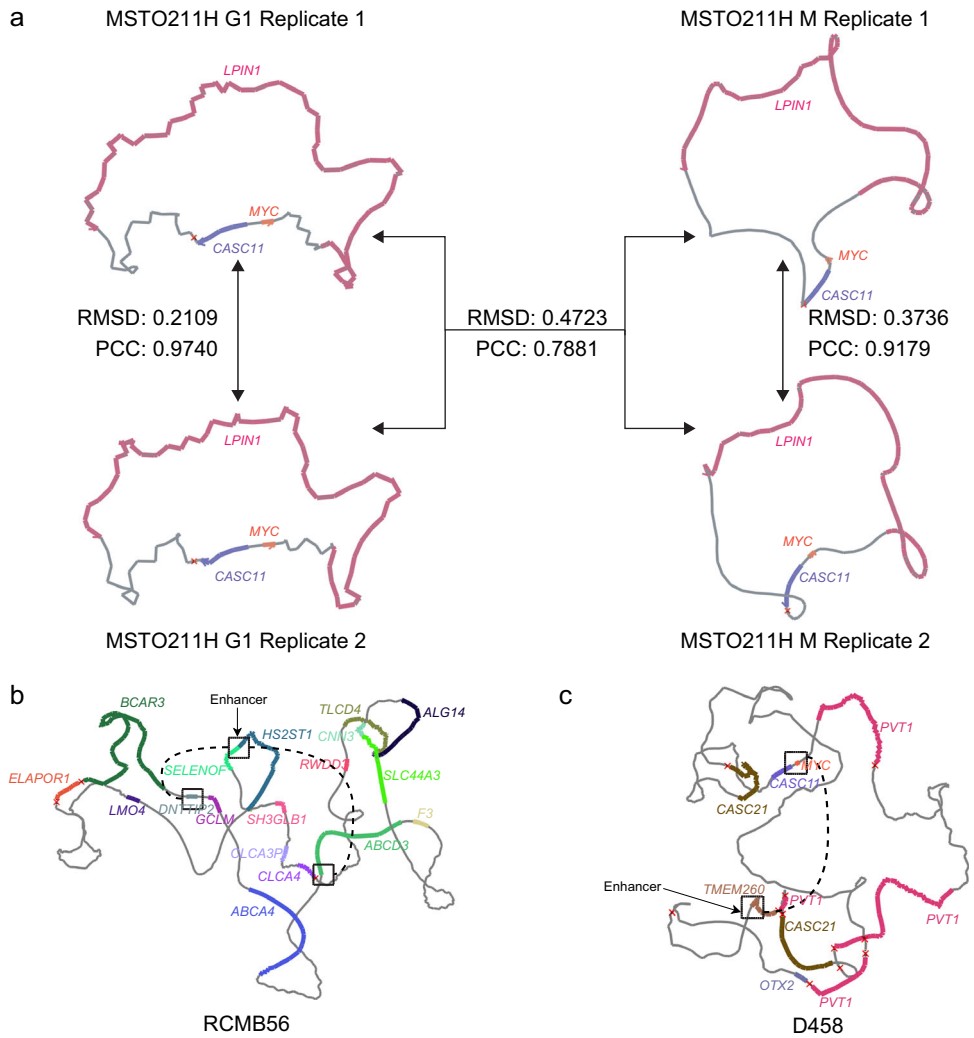

**Fig. 5 | Characteristics of ecDNA conformation during cell cycle and hub formation. a** Two Hi-C replicates were performed for each cell cycle phase (G1 and M). Ensembles of 5 locally optimal 3D structures were generated from each Hi-C replicate. RMSD and PCC values were averaged over all combinations of ensemble structures. **b** Differential circ-SI interactions that are spatially distant, suggesting

potential *trans* (ecDNA-ecDNA) interactions in RCMB56. Both interactions connect an active enhancer at *SELENOF* locus to remote oncogenes (*DNTTIP2* and *ABCD3*). **c** Differential interactions in D458 connecting *MYC* and a remote enhancer on chr14. Source data are provided as a Source Data file.

captured interactions that could be directly attributed to higher spatial proximity in the ec3D reconstruction, relative to their genomic distance on the ecDNA. Thus, circ-SI and spatial-SI captured decreasing subsets of the interactions predicted by ref-SI.

The number of ref-SI interactions in the ecDNA of GBM39 and RCMB56 was significantly larger relative to the identical region in controls GM12878 and IMR90, and the difference was most pronounced at larger genomic distances due to circularization (two-sample Kolmogorov-Smirnov test p-values: GBM39-GM12878 = 0.024, GBM39-IMR90 = 0.00016, RCMB56-GM12878 = 1.2e-36, RCMB56-IMR90 = 1.1e-71; Supplementary Fig. 21). The results did not change even after rescaling the control matrices to correct for the higher copy number of ecDNA (**Methods**). Furthermore, the proportion of distance-dependent significant interactions remained consistent despite variation in the total number of interactions.

## Ec3D predicts interactions that corroborate heterogeneity of ecDNA sequence

Sequence heterogeneity exists when different copies of ecDNA do not have identical sequences. It is observed when the dominant cycle identified by sequence reconstruction tools, such as

AmpliconArchitect (AA) or CoRAL, does not explain all of the copy number change[20]. This information is known to ec3D, which uses the dominant cycle to generate a 3D structure and identifies significant interactions. All ecDNA-positive cell lines except TR14 carried a single dominating ecDNA species. A CoRAL reconstruction of a TR14 ecDNA carrying *MYCN* suggested 3 overlapping cycles with estimated copy numbers 279 (explaining 65.7% of the amplification), 86 (20.2%), and 30 (Supplementary Fig. 22a). We analyzed ec3D predictions of these ecDNA cycles to investigate sequence heterogeneity. Ec3D-derived structure using Cycle 1 (Supplementary Fig. 22b) revealed many significant interactions. Some were mediated by 3D topological constrictions (Supplementary Fig. 22b, c). In contrast, the interactions marked SV1-3 were not supported by the ec3D conformation. However, they were proximal to breakpoints created by Cycle 2, thereby independently corroborating its existence. Cycle 3 has a much lower copy number that explains only 5.8% of the amplicon, causing the reduced strength of the SV4 interaction.

## Ec3D reconstruction captures "crossing" interactions

We next investigated if ecDNA could have significant interactions with a non-planar topology, unlike interactions on TADs that are

represented as diagonal blocks consistent with a planar topology. One discernible feature of a complex or non-planar 3D fold is the presence of crossing significant interactions, which can be described by 4 remote loci, or two interacting pairs $(x, z)$ and $(y, w)$, such that $x < y < z < w$, corresponding to two topological constrictions (Supplementary Fig. 23a). While the smaller ecDNA structures (e.g., GBM39, Fig. 3b) encompassed only a single topological constriction, other, larger ecDNAs contained multiple topological constrictions with crossing interactions (Supplementary Data 3). For example, we identified 15 crossing interactions from the D458 ecDNA[17]. Among these interactions, one between 500 kb distal sites on the *MYC-PVT1* locus crossed another interaction that connected the region upstream of *TMEM260* in chromosome 14 with a region upstream of *CASC8* on chromosome 8 (Supplementary Fig. 23b). The results suggest that ecDNA can promote novel interactions utilizing not only structural variation but also complex topological constrictions.

### Ec3D reconstructions identify multi-way interactions

We used Louvain clustering (**Methods**) to obtain clusters of ref-SI interactions for each of the cell lines, suggestive of complex regulatory networks. In D458, we identified 6 ref-SI clusters (Supplementary Data 9). One of these was a clique-like interaction among multiple loci on chr8 and chr14: chr8:127.95-128.02 Mb, the *PVT1* locus; chr8:128.44-128.58 Mb; chr8:128.70-128.74 Mb; and chr14:56.80-56.88 Mb, the *OTX2* locus (Supplementary Fig. 24a). A second cluster (cluster 4) from the same cell line showed a star-like connectivity where a central region containing *MYC*, *PVT1* interacted with multiple distal loci situated ~430 kb 5′ upstream, and ~800 kb, ~1.01 Mb, and ~1.06 Mb 3′ downstream of the *MYC/PVT1* region (Supplementary Fig. 24b). The regions upstream and downstream of MYC/PVT1 are devoid of coding genes but contain ncRNA including *CASC8* and *CASC21*. These findings support earlier studies that show the co-amplification and ecDNA formation of these two distinct regions in multiple acute myeloid leukemia samples[48]. It is also notable that the ncRNA *PVT1* appears (partially) with 4 copies in the ecDNA with SV driven proximity to *OTX2*, *TMEM260* (natively on chr14), and *CASC8*, *MYC* (chr8) consistent with its role in mediating gene fusions[49].

### Ec3D reveals differential Hi-C interactions

As described earlier, we used circ-SI to identify significant interactions that cannot be attributed to structural variations. We next used spatial-SI to identify significant interactions that were specifically due to spatial proximity. Indeed, in the simpler structures such as GBM39, interactions in circ-SI were also identified using spatial-SI (Supplementary Fig. 25). Surprisingly, in the ecDNA of RCMB56 (Supplementary Fig. 26) and D458 (Supplementary Fig. 27), we observed many *differential* interactions – interactions in circ-SI that were not identified by spatial-SI. These interactions could not be attributed either to sequence proximity created by structural variation in ecDNA or to spatial proximity corresponding to topological constrictions. We also did not find evidence of other structural variations that could indicate heterogeneity of ecDNA in the sample.

Many mechanistic reasons could explain these differential interactions. They could, for example, occur due to heterogeneity of ecDNA structure. Another intriguing hypothesis is that these differential interactions are *trans interactions*, where regulatory elements in one ecDNA are utilized by a different ecDNA in the same hub, as has been suggested previously[14,50].

We explored the occurrence of known regulatory sites in regions with differential interactions. In RCMB56, the region from chr1:86.905-86.935 Mb interacted with multiple distal regions, including chr1:93.845-93.885 Mb, containing the oncogene *DNTTIP2*, and chr1:94.410-94.430 Mb, containing *ABCD3* (Fig. 5b). An H3K27Ac peak, reflective of an active enhancer, was prominent at chr1:86.915 Mb in multiple tissue types[51] (Supplementary Fig. 26). Similarly, we observed

a multi-chromosomal trans interaction between chr8:127.73-chr8:127.745 Mb and chr14:56.645-56.675 Mb in D458 (Fig. 5c), where the chr8 region contained the oncogene *MYC*, and the chr14 region contained an active enhancer mark (Supplementary Fig. 27).

## Discussion

EcDNAs are circular acentric molecules that are exclusively and ubiquitously found in cancer cells, where they are responsible for oncogene amplification and increased pathogenicity. Their unusual shape and highly accessible chromatin allow for enhancer hijacking and regulatory rewiring. Here, we add another layer of understanding of ecDNA by presenting an algorithm to reconstruct its 3-dimensional structure using chromatin capture data.

While these are large molecules and not expected to have a rigid structure, our results on extensive simulation experiments and on real data suggest that most proximities are accurately captured by ec3D. Larger $\alpha$ values imply relatively stronger interactions even between spatially distant regions, adding more information for our structure reconstruction. Because ecDNAs are formed by joining multiple distinct genomic segments and are circular, Hi-C interaction is exactly strong between a pair of bins when they are brought proximal either because of the structural variation or because of a topological constriction. In all experiments, ec3D reconstructions consistently showed strong inverse correlations between the spatial distance of bin pairs and the strength of their Hi-C interactions.

Compared with *Multidimensional Scaling* (MDS) based methods[31,35,52,53], which attempt to minimize a stress function that measures a discrepancy between the "wish distances" and the 3D distances of the structure, the Poisson model[32] allows more flexible handling of duplicated segments, as one either has to compute wish distances between each copy of a duplicated bin and other bins by splitting the interactions; or introduce a stress function to measure the discrepancy between the expected and observed interactions in duplicated regions.

The challenge of 3D reconstruction of DNA structures can potentially be addressed by other complementary methodologies. Multiplexed imaging of hundreds of genomic loci by sequential hybridization has the potential to elucidate 3-dimensional structures of entire chromosomes at single-cell levels, albeit typically with a tradeoff between throughput and genomic resolution[39,40,54]. We therefore focused on chromatin capture data due to finer genomic resolution and ease of data acquisition, although the data was acquired from bulk and not from single cells. We also acquired super-resolution imaging data at 200-kb genomic resolution to compare against ec3D structures. The results, averaged over 10 ecDNAs, revealed a strong correlation in the pairwise distances predicted by imaging and ec3D. In future experiments, we plan to further utilize multiplexed imaging with finer genomic resolution[38–40], to better understand ecDNA structure at the single-cell level.

Ec3D requires an assembled ecDNA sequence to work with. We designed it to directly utilize the outputs of AmpliconArchitect or CoRAL, which use whole-genome sequencing to automatically identify ecDNA and reconstruct plausible sequences. Thus, ec3D cannot be used without accompanying whole-genome sequencing. Although Hi-C technology has been used to detect structural variation and 3D conformation, reliable ecDNA discovery currently requires WGS data. In future development, we will focus on methods that detect ecDNA and resolve its sequence and structure using Hi-C data.

When counterpart WGS is available, tools like AmpliconArchitect and CoRAL often identify one dominant cycle that explains most of the observed copy number and structural variation. In this case, the predicted sequence (generated by traversing the cycle) is likely to be correct, and ec3D would work directly. In many of our analyzed samples, sequence heterogeneity was indeed low, and a single sequence explained the entire copy number. In other cases, multiple ecDNA

species with varying sequences may exist. In these cases, ec3D provides the structure of a plausible ecDNA using the dominant cycle but may also reveal evidence of sequence heterogeneity. In TR14, ec3D revealed sequence heterogeneity by identifying significant interactions near structural variants that were not part of the dominant cycle. Moreover, the strength of the interactions correlated with the copy numbers of the alternative ecDNA sequences. Ec3D can also provide information about the quality of ecDNA reconstruction by analyzing correlations between the ec3D-predicted pairwise distances and the Hi-C-generated interaction frequencies. Recently, Zhao et al.[55] have utilized ec3D to support the existence of two different ecDNA species in a Glioblastoma tumor sample. In future work, we also plan to develop an integrated tool for identifying the ecDNA sequence using both WGS and Hi-C data. When multiple distinct ecDNA species exist, novel methods will be required to deconvolute the Hi-C data to provide a consensus structure for each species.

The individual ecDNA molecules with the same sequence may additionally have slightly different structures. In the presence of this structural heterogeneity, we focused on providing one representative (i.e., *consensus*) structure. We observed a strong negative correlation between the number of Hi-C interactions in a pair of bins and their predicted spatial distance, across all molecules tested, providing confidence in the predicted consensus structure. The consensus structure can be used to generate hypotheses, design experiments, and integration with other data types (e.g., ChIP-seq). Furthermore, consensus structures have proven useful in revealing the components of regulatory machinery in a region, including topologically associating domains (TADs) and chromatin loops[8]. While consensus methods are commonly used in 3D reconstruction, there are also ensemble methods for reconstruction, and ensemble structures provide complementary information. We briefly investigated ensembles by generating multiple locally optimal structures for the MSTO211H cell line. The results revealed that ensemble structures within the same biological condition (G1 or M) were highly similar to one another. In contrast, structures derived from G1 differed substantially from those in M phase. Thus, consensus structures provide a reliable representation of ensembles while also capturing relevant differences in different cell cycle phases. Additionally, ecDNA structures are governed by external influences such as different ecDNA-protein interactions and ecDNA-chromosome tethering. Future work will focus on the mechanistic underpinnings of these structural changes.

Because ecDNAs are large molecules with the flexibility of DNA conformations, we hypothesized that their three-dimensional structure was not entirely intrinsic but was impacted by interactions with proteins, including those involved in gene regulation. It had been shown previously that ecDNAs generate new topologically associated domains and rewire the regulatory circuitry with previously inaccessible enhancer regions hijacked by oncogenes. To test this phenomenon more, we first looked at the volume occupied by ecDNA. Our results suggest that ecDNAs fully occupy a 3-dimensional volume, making their shape less disk-like and more oblate spheroidal. The 3-dimensional shape allows for more complex patterns of interaction and possibly rewires the regulatory circuitry in ways that could be quite different from the chromosome. Indeed, our analysis of significant interactions revealed many interesting cases; we found "crossing" interactions which would not be possible in a planar structure; examples of clique-like and star-like interactions implying proximity of multiple regions (multiple enhancer elements regulating a gene); and also possible evidence of *trans* interactions between different ecDNA molecules. These early findings provide hypotheses that can be tested in future work, for example, through changes in differential interactions upon dissociation of ecDNA hubs.

We used ec3D to investigate the structure of amplified regions on isogenic lines which were mostly identical except for the location of focally amplified regions, which are either extrachromosomal or intrachromosomal. Remarkably, the amplicon had very similar structures, suggestive of similar regulatory patterns. Indeed, in addition to neo-TADs, chromosomal TADs have also been observed on ecDNA. These reconstruction data also shed light on the possible 3-dimensional structure of HSRs. As the resolution of Hi-C data improves, we can use our methods to better distinguish between different HSR configurations.

The ec3D algorithm can work even when the ecDNA contains duplicated segments whose interactions are all collapsed in the input Hi-C data. We investigated the fine structure of duplicated regions and found that while some duplicated regions have very similar structures, others do not, consistent with the idea that the 3D structure of ecDNA is not intrinsic to its sequence but is mediated by interacting proteins. We even found a significantly similar structure of the same region amplified in two different cell lines, suggesting common patterns of regulatory wiring across different samples.

There are many future avenues for improving methodology. Clearly, the technology requires a complete and correct ecDNA primary sequence. This is often challenging with short-read-based reconstruction, which could be ambiguous and miss many critical breakpoints. Here, we selected ecDNA sequences that were tractable and showed minimal cell-to-cell heterogeneity. This method should not be deployed straight out of the box into unvalidated sequences. Patient samples, where there is often greater heterogeneity than in cancer cell lines, should be analyzed carefully for interactions that correspond to structural variation from heterogeneous species. The genomic analysis of complex and heterogeneous ecDNAs will improve with long-read technologies[20], making it easier to utilize ec3D.

Recent methods for single-cell Hi-C[56,57] will allow for measurements of cell-to-cell variability of ecDNA structures and also help elucidate the structures of multiple ecDNAs in the same sample. Methods are also being developed that disrupt the tethering of ecDNA to chromosomes or to other ecDNA[14]. Future work aimed at studying the change in structure due to the disruption of tethering could help identify the DNA elements involved in tethering, resolving an important biological problem.

In summary, ec3D provides a tool for the exploration of the regulatory biology of extrachromosomal DNA and other focal amplifications.

## Methods

### Modeling genomic duplications in Hi-C

The input ecDNA genome often contains duplicated segments of a reference genome. Standard Hi-C mapping and binning methods are unable to separate the interactions on (and between) each distinct copy of a duplicated segment; instead, we observe the sum of interactions given by all copies of that segment. Formally, we refer to the collapsed matrix as the Hi-C matrix, where each duplicated segment occurs only once; and the expanded matrix as the Hi-C matrix representing the structure of ecDNA, where all duplicated segments occur as many times as they are duplicated. Note that only the collapsed matrix is observed. The expanded matrix, which must be inferred, determines the structure of ecDNA and the significant interactions on ecDNA.

To differentiate collapsed matrices and expanded matrices, we use the following notations throughout the method description:
- $N_e$: total number of fixed resolution bins in the expanded matrix. We typically use 5 kb. The size of the expanded Hi-C matrix is $N_e \times N_e$.
- $N_c$: total number of bins involved in ecDNA in the collapsed Hi-C matrix, which is of size $N_c \times N_c$.
- $L_i$: denotes the genomic coordinates (at 5 K resolution) corresponding to bin $i$. Note that in an expanded matrix, different bins may have the same genomic location if they come from duplicated segments on ecDNA.

- $\mathcal{R}_i$: For each bin $i$ in the collapsed matrix, $\mathcal{R}_i = \{a \in \{1, \cdots, N_e\} | L_i = L_a\}$ denotes the set of indices in the expanded Hi-C matrix that have the same genomic location as bin $i$. The bin $i$ is denoted as *unique*, if $|R_i| = 1$, and *duplicated* otherwise.
- $C_{ij}$: #interactions between bins $i, j$ in the collapsed Hi-C matrix.
- $E_{ab}$: #interactions between bins $a, b$ in the expanded Hi-C matrix.

We make the assumption that the observed number of interactions $C_{ij}$ between a pair of bins $i, j$ is given by:

$$C_{ij} = \sum_{a \in \mathcal{R}_i} \sum_{b \in \mathcal{R}_j} E_{ab}. \tag{1}$$

The following methods are developed based on this principle.

## Preparing ecDNA Hi-C matrices

Ec3D's three-dimensional reconstruction only depends on interactions within the ecDNA intervals. Therefore, we first create an ecDNA Hi-C matrix by extracting, reassembling and reorienting the submatrices corresponding to interactions between pairs of segments composing the ecDNA. The input ecDNA sequence is given as a list $S = [(s_1, o_1), (s_2, o_2), (s_3, o_3), \cdots]$ of ordered and oriented genomic segments, where each $s_i$ denotes a genomic interval and $o_i \in \{'+', '-'\}$ indicates the orientation of $s_i$. The Hi-C data is provided as a matrix of interactions between genomic bins from the whole genome. As a first step, we map each segment to a collection of bins, allowing for duplications, to obtain the $N_e$ bins that are amplified by the ecDNA. For each pair of segments $(s_i, s_j)$, we extract the corresponding submatrix of binned Hi-C interactions, and reassemble these submatrices into a single matrix $E$ of size $N_e * N_e$ bins according to their order in $S$, with inverted segments ($o_i = '-'$) reoriented (Fig. 1). Next, we iteratively remove all rows (and columns) $a'$ if there exists a column $a < a'$ with $L_{a'} = L_a$ in $E$. This results in a collapsed matrix $C$, to be used subsequently. Additionally, we keep the mapping of indices from the expanded matrix $E$ to the collapsed matrix $C$ to query the indices in each $\mathcal{R}_i$.

## Normalizing ecDNA Hi-C matrices

The Hi-C data is typically normalized, for example, using ICE normalization[58], to correct for bin-to-bin variation by ensuring that for each bin $i$ in the normalized matrix $C^{ICE}$, $\sum_j C^{ICE}_{ij} = 1$.

Within the ecDNA Hi-C matrix, we also ignore the copy numbers contributed by the normal chromosomes as they are much smaller than the ecDNA copy numbers, and copy numbers are uniform across the ecDNA. However, normalization must account for duplications of genomic regions within the ecDNA. With an expanded matrix $E$, we could enforce $\sum_a E^{ICE}_{ab} = 1$. Instead, we work directly with the collapsed matrix, and aim to compute $C^{ICE}_{ij} = \sum_{a \in \mathcal{R}_i} \sum_{b \in \mathcal{R}_j} E^{ICE}_{ab}$. But since $E^{ICE}$ is not known, we approximated $C^{ICE}_{ij}$ through a generalized version of ICE normalization such that in the normalized matrix $C^{ICE}$ (of the reassembled matrix $C$), $\sum_i C^{ICE}_{ij} = |\mathcal{R}_i|$, where $|\mathcal{R}_i|$ is the multiplicity of genomic bin $i$ on ecDNA. Finally, to keep the original scale of interactions, we multiply a constant $r = (\sum_i \sum_j C_{ij})/N_c$ to the normalized matrix $C^{ICE}$ and work on the scaled matrix $r \cdot C^{ICE}$ in the following steps. We implemented the normalization procedure above using the iced package[59].

## Reconstructing the 3D structure of ecDNA

Given a normalized Hi-C matrix for ecDNA $C^{ICE}$ (or $r \cdot C^{ICE}$), we compute a single consensus (of multiple copies of ecDNA in a mixture of cells) 3D structure. Formally, we compute a vector $X \in \mathbb{R}^{N_e \times 3}$ of dimension $N_e \times 3$ - where $X_a = (x_{a1}, x_{a2}, x_{a3})$ represents the

coordinate of bin $a \, (a \in 1, \cdots, N_e)$. Define

$$d_{ab} = ||X_a - X_b||_2 = \sqrt{(x_{a1} - x_{b1})^2 + (x_{a2} - x_{b2})^2 + (x_{a3} - x_{b3})^2}$$

as the Euclidean distance between bin $a$ and bin $b$ given the coordinates of $X_a$ and $X_b$.

The normalized interaction frequency $C^{ICE}_{ij}$ is modeled as a Poisson random variable, relating to $d_{ij}$. Specifically, for a pair of unique bins $i, j$, the expected number of interactions is given by $\lambda_{ij} = \mathbb{E}[C^{ICE}_{ij}] = \beta d^{\alpha}_{ij}$, for parameters $\alpha < 0, \beta > 0$, which are estimated separately for each dataset. The parameter $\alpha$ describes the rate of power law decay of Hi-C interactions due to spatial distances, and $\beta$ can be treated as a scaling factor. Moreover, the likelihood of observing $C^{ICE}_{ij}$ interactions between a pair of bins $i, j$ is given by a Poisson(-like) distribution:

$$\mathcal{L}(C^{ICE}_{ij}, X) = \frac{\left(\lambda_{ij}\right)^{C^{ICE}_{ij}} \exp\left(-\lambda_{ij}\right)}{\Gamma(C^{ICE}_{ij} + 1)} \tag{2}$$

When the bin pairs are not unique, we define $\lambda_{ij} = \mathbb{E}[C^{ICE}_{ij}] = \sum_{a \in \mathcal{R}_i} \sum_{b \in \mathcal{R}_j} \beta d^{\alpha}_{ab}$, and the likelihood is computed based on the new expectations.

We aim to maximize the log likelihood of the overall collapsed matrix $C^{ICE}$ or minimize $-\ln(\prod_{i,j} \mathcal{L}(C^{ICE}_{ij}, X))$. Additionally, since ecDNA forms a continuous polymer chain with fixed resolution bins as discrete points along the chain, we introduce a regularization term to control the variance between consecutive bins $a$ and $a + 1$:

$$Reg(X) = Var(d_{a, a+1}) = \frac{1}{N_e - 1}\left(\sum_{a=1}^{N_e-1} d^2_{a, a+1} - \left(\sum_{a=1}^{N_e-1} d_{a, a+1}\right)^2\right) \tag{3}$$

This regularization term ensures equal spacing of consecutive bins in the Euclidean space. The overall optimization problem is then given as follows

$$\min -\ln\left(\mathcal{L}(C^{ICE}, X)\right) + \gamma \cdot Reg(X) \tag{4}$$

or equivalently

$$\min \sum_i \sum_j \left(\lambda_{ij} - C^{ICE}_{ij} \ln\left(\lambda_{ij}\right)\right) + \frac{\gamma}{N_e - 1}\left(\sum_{a=1}^{N_e-1} d^2_{a, a+1} - \left(\sum_{a=1}^{N_e-1} d_{a, a+1}\right)^2\right) \tag{5}$$

with a constant term not depending on $X$ ignored in the minimization. The weight $\gamma$ of the regularization term is provided as a user input, and by default we set $\gamma$ to $0.05 \cdot N_e$.

## Implementation details

The optimization is done iteratively, for $X$ and $\alpha$ (and $\beta$), with l-BFGS[60] algorithm implemented in SciPy:

1. Start with an initial estimation of $X$;
2. Minimize the negative log likelihood with respect to $\alpha$ and $\beta$ by fixing $X$;
3. Minimize the negative log likelihood over $X$ after fixing $\alpha$ and $\beta$;
4. Iterate steps 2 and 3 until convergence or reaching an upper bound of rounds (by default we set the maximum number of rounds to 1000).

To determine convergence, we look at the value of objective function in the last 10 rounds and set the convergence criteria to $|obj_i - obj_{i-10}| / \max(obj_i, \cdots, obj_{i-10}) < \epsilon$, where $obj_i$ and $obj_{i-10}$ are

objective values at the current round and 10 rounds before, respectively. To avoid local minima due to non-convexity, we run the initialization and iterative optimization multiple (5 by default) times, with random initialization of $X$ for running MDS (see below for the initialization of $X$). The user can choose to output the final $X$ that leads to the best objective value as a consensus structure, or all final structures as an ensemble of structures. In the optimization process, we require that the three dimensions are bounded by [-1, 1], but do not enforce any limit on $\beta$ to allow flexible scaling of the structures.

## Initialization of $X$

We found that initialization plays an important role in deriving the optimal coordinates $X$ (**Results**, Supplementary Fig. 8), and hence we try to initialize $X$ sufficiently close to the final solution by initializing $X$ with running a procedure similar to multidimensional scaling (MDS)[32].

Note that the naive MDS requires the expanded matrix to work with. To obtain the expanded matrix for MDS, we redistribute the normalized interactions $C_{ij}^{ICE}$ to $E_{ab}$ for all $a \in \mathcal{R}_i, b \in \mathcal{R}_j$ in proportion to $d_{ab}^{-3}$ (i.e., with the assumption that $\alpha = -3$; $\beta$, the scaling factor, can be canceled out here). Thus,

$$E_{ab} = \frac{d_{ab}^{-3}}{\sum\limits_{a' \in \mathcal{R}_i} \sum\limits_{b' \in \mathcal{R}_j} d_{a'b'}^{-3}} \cdot C_{ij}^{ICE} \text{ if } i \neq j. \tag{6}$$

When $a, b \in \mathcal{R}_i$, we set

$$E_{ab} = \text{Avg}(C_{i'j' : \mathcal{R}_{i'} = \{a'\}, \mathcal{R}_{j'} = \{b'\}, d_{a'b'} = d_{ab}}^{ICE}) \tag{7}$$

(i.e., the average of all unique bin pairs $a'$ and $b'$ with the same distance as bin pair $a, b$; and we use genomic distance as defined below) when redistributing the diagonal elements. Since the Euclidean distance is not known, we use a *circular* genomic distance on ecDNA as a proxy: $d_{ab} = g_{ab} = \min(|a - b|, N_e - |a - b|)$, i.e., the shortest distance between bin $a$ and $b$ on the circular ecDNA structure. To better compute $X$ we allow some flexibility in redistributing interactions by treating $E_{ab}$ as variables in the optimization process and adding a stress function to penalize the discrepancies between $\sum_{a \in \mathcal{R}_i} \sum_{b \in \mathcal{R}_j} E_{ab}$ and $C_{ij}$.

Specifically, the objective of MDS can be written as

$$\min \sum_{a=1}^{N_e} \sum_{b=1}^{N_e} (d_{ab} - \delta_{ab})^2 / \delta_{ab}^2 + \sum_{i=1}^{N_c} \sum_{j=1}^{N_c} \left( \sum_{a \in \mathcal{R}_i} \sum_{b \in \mathcal{R}_j} E_{ab} - C_{ij}^{ICE} \right)^2 / C_{ij}^{ICE} \tag{8}$$

where $\delta_{ab} = (E_{ab}/\beta)^{-1/3}$ is the wish distance. Again, we set $\alpha = -3$ regardless of the true/optimal values as MDS is run just for initialization purposes.

## Resolving HSRs created through reintegration of ecDNA

We preprocessed data to reconstruct the structure of HSRs formed by head-to-tail recombination of the ecDNA sequence and subsequent chromosomal reintegration. We ran CoRAL[20] to obtain a single copy composing this underlying tandem-duplication like HSR genome (see section **ecDNA genome reconstruction from WGS data** below) and duplicated the first 3 bins, representing 15 kb of sequence during preparation of the collapsed matrix. The predicted CoRAL sequence along with the 15 kb duplication was provided as input to ec3D for structure reconstruction. Ec3D automatically normalized the collapsed matrix and reconstructed HSR structures.

## Identifying significant interactions

Increased Hi-C interactions can be attributed to three main factors: (a) reference genome proximity, which leads to spatial proximity, (b)

spatial proximity induced by structural variants (SVs)[28,61–63], and (c) spatial proximity introduced by a conformational change. Furthermore, due to the higher copy number of ecDNA and potential formation[14,50], significant interactions may also reveal *trans* interactions between two ecDNA molecules. As per previous methods[25–27], we define significant interactions as pairs of bins $(a, b)$ $(a, b = 1, \cdots, N_e)$ with interaction frequencies $E_{ab}$ much more than expected at a given genomic distance. We first introduce a unified method in ec3D that computes significant interactions for an abstract definition of genomic distance here. In the next subsection, we describe different choices of genomic distance functions that allow us to distinguish interactions due to SVs from interactions due to conformational change.

Specifically, to identify statistically significant interactions, we always model the interaction frequencies at each genomic distance $g$ using a negative binomial distribution with mean $\mu_g$ and variance $\sigma_g$ ($\sigma_g > \mu_g$). The statistical significance (p-value) of $E_{ab}$ is computed as the probability of observing at least $E_{ab}$ interactions with the underlying distribution: $P_{ab} = \mathbb{P}(e \geq E_{ab}), e \sim NB(\mu_g, \sigma_g), \forall a, b$ satisfying $g_{ab} = g$. Then we correct all resulting p-values for multiple testing using the Benjamini-Hochberg procedure to compute an adjusted p-value (i.e., q value) for each bin pair $(a, b)$. By default, pairs of bins with q value $< 0.05$ are denoted as significant interactions. We noticed that significant interactions often occurred clumped with their neighboring bin pairs in the Hi-C matrix, at high resolutions such as 5 K. Therefore, we implemented an option to only output the locally maximal significant interactions, i.e., those with interaction frequencies greater than their top, bottom, left, and right neighbors.

The mean ($\mu_g$) and variance ($\sigma_g$) of the number of interactions at each genomic distance $g$ are estimated by computing the empirical mean and variance of interactions $E_{ab}$ for all $a, b$ satisfying $g_{ab} = g$, after detecting and removing outliers using the IQR method[64].

The computation of significant interactions also requires the expanded matrix $E_{ab}$. To compute the expanded matrix, we first redistribute raw interactions $C_{ij}$ to $E_{ab}$ for all $a \in \mathcal{R}_i, b \in \mathcal{R}_j$ similar to Eqs. (6) and (7) described in **Initialization of $X$**, but using the optimal values of spatial distance $d_{ab}$ and $\alpha$:

$$E_{ab} = \frac{d_{ab}^{\alpha}}{\sum\limits_{a' \in \mathcal{R}_i} \sum\limits_{b' \in \mathcal{R}_j} d_{a'b'}^{\alpha}} \cdot C_{ij} \text{ if } i \neq j;$$

and

$$E_{ab} = \text{Avg}(C_{i'j' : \mathcal{R}_{i'} = \{a'\}, \mathcal{R}_{j'} = \{b'\}, d_{a'b'} = d_{ab}})$$

otherwise. The resulting expanded matrix $E_{ab}$ is then renormalized with ICE normalization.

Finally, we exclude potential false positive calls due to an artifact of ICE normalization. For example, in RCMB56 matrix, row (or column) 27 has only a few non-zero entries, potentially due to a mapping/binning artifact of HiC-pro, but ICE normalization forces this row to have the same sum of interactions as other rows. As such, the interaction counts were boosted by ICE normalization and returned as significant in the p-value calculation. We postprocess significant interactions by removing all rows with far fewer non-zero entries than the average, again with the IQR method. Ec3d implements a user option to remove interactions in certain rows/columns.

## Choosing genomic distance between two bins

The circular genomic distance $g_{ab}^c = \min(|a - b|, N_e - |a - b|)$ defined in section **Initialization of $X$** implicitly removes the effect of SV breakpoints joining remote genomic segments (in $S$) on ecDNA. In contrast, to capture both SV-driven and conformation-driven significant interactions, we define the genomic distance between bins $a$

and $b$ as their genomic distance on the reference genome:

$$g_{ab}^r = \min(|L_a - L_b|, g_{max}) \qquad (9)$$

when $L_a$ and $L_b$ are located on the same chromosome, where $g_{max}$ is a sufficiently large genomic distance with no (or few) interactions on expectation at this distance; and $g_{ab} = g_{max}$ when $L_a$ and $L_b$ are located on different chromosomes. Notice that genomic distance is not continuous in Hi-C - two adjacent values differ by a fixed resolution, e.g., 5 kb. We set $g_{max}$ to the size of ecDNA to avoid large gaps between two genomic distances. We refer to **ref-SI** as bin-pairs $(a, b)$ where the number of Hi-C interactions between loci in bins $a$ and $b$ was significantly higher than expected for the reference genomic distance $g_{ab}^r$ between the bins; **circ-SI** as pairs $(a, b)$ where the number of Hi-C interactions was significantly higher than expected for the *circular* distance $g_{ab}^c$.

The number of bin pairs on ecDNA usually decreases (linearly) with increasing reference genomic distance. Bin pairs that do not share the same genomic distance with any other pairs are unlikely to be identified as significant, due to the way p-values are calculated. Therefore, we sort all bin pairs according to their genomic distances, and partition them into groups, each with at least $\frac{N_e}{2}$ bin pairs, by greedily merging bin pairs that are similar in genomic distance. The mean and variance of the number of interactions are estimated separately for each group and used to compute nominal p-values. If circular genomic distance is used, this partition is not needed as the number of bin pairs at each genomic distance remains the same (i.e., $N_e$), except for a maximum distance with $\frac{N_e}{2}$ bin pairs when $N_e$ is even.

## Identifying candidate *trans* interactions

Earlier research has revealed that ecDNA forms hubs with regulatory interactions between different ecDNA molecules[14,50]. Therefore, it is possible that ecDNA Hi-C data includes interactions between distinct copies of ecDNA molecules. To identify *cis* interactions within ecDNA, we can optionally compute significant interactions with respect to the ratio $g_{ab}^c / d_{ab}$ between circular genomic distances and spatial distances. Specifically, for each distance $g$, and all bin pairs $(a, b)$ such that $g_{ab}^c = g$, we fit a negative binomial distribution for the ratio $g / d_{ab}$. Pairs of bins with significantly high ratio after FDR corrections corresponded to significant interactions relative to their spatial proximity. We refer to this third measure of significant interactions as **spatial-SI**. Significant interactions computed from Hi-C using circ-SI that are not found using spatial-SI are suggestive of "secondary" interactions. These interactions can result from alternative 3D conformations, secondary SVs (not participating in the ecDNA sequence), or *trans* interactions between ecDNAs.

## Identifying significant interactions from rescaled matrices

To show that significant interactions on ecDNA were not due to their higher copy numbers, we rescaled the case (i.e., ecDNA) and control (i.e., extracted from the same intervals from non-amplified cell lines) Hi-C matrices by a factor ranging from 0.25 to 4, and then identified significant interactions with the same procedure but from the rescaled matrices. The results suggested that the number of significant interactions is not monotonically increasing with the total number of interactions, and the pattern of significant interactions as a function of increased genomic distance remains the same. In fact, the number of significant interactions reached a local maximum in most cases without rescaling. We note that the variance of interactions at each genomic distance decreased quadratically with the downscale factor, breaking the negative binomial property when the rescaling factor becomes too small. As such, we only tested rescaling factors that preserve larger variance than mean interactions at 90% of all distinct genomic distances.

## Clustering significant interactions

EcDNA often exhibits complex conformations that form multi-way interactions among different regions within its structure to amplify the oncogene and other associated gene expression. The connectivity of these multi-way interactions (e.g., star-like shape or clique-like) indicates different types of interacting pathways. To identify multi-way interactions, we build an interaction network $G = (\mathcal{V}, \mathcal{E})$ from all significant interactions where the node set $\mathcal{V}$ includes all bins involved in a significant interaction and the edge set $\mathcal{E}$ indicates the actual interactions. We detect *communities* in the interaction network $G$ by using Louvain clustering. Louvain clustering[37] partitions nodes into clusters while maximizing the modularity score (density of links within clusters compared to links between clusters).

## Simulations

We simulated ecDNA 3D structures and their corresponding Hi-C data to assess the effectiveness of ec3D in 3D structure reconstruction. At the highest level, we introduced the notion of *topological constrictions* to simulate the effect of major conformational changes on ecDNA structures. Topological constrictions generalize chromatin loops - which typically connect a pair of bins (x, y) that are genomically far - by specifying two broader intervals of bins $[x, x + \Delta x]$, $[y, y + \Delta y]$ where the neighboring bins around x and y are generally genomically distant but spatially close, resulting in strong off-diagonal Hi-C interactions. An increased number of topological constrictions usually indicates more complex 3D structures.

Each simulated structure was obtained by sampling evenly spaced points from a circular 3D curve. We generated a diverse set of *base structures* by varying three key parameters, which determine the shape of the underlying 3D curve and the number of points to be sampled. First, we incorporated $k \in \{1, 2, 3\}$ topological constrictions. Second, we varied the spatial distance between the two intervals that participate in a topological constriction. Third, we simulated structures of different sizes by varying the total number of points $N_e \in \{250, 500, 750\}$. In addition, we introduced *local folds* on each base structure by randomly disturbing the positions of small collections of continuous points. See Supplementary Fig. 1 and **Supplementary Methods** for details. In these simulated structures, each point can be treated as the spatial placement of a genomic bin at a fixed Hi-C resolution.

We next generated an expanded Hi-C matrix $E$ of size $N_e \times N_e$ from each simulated circular structure with $N_e$ bins as follows. For each pair of bins $a, b \in \{1, 2, \cdots, N_e\}$, we sampled the interaction counts $E_{ab}$ from a Poisson distribution with mean $\beta d_{ab}^{\alpha}$[30], where $d_{ab}$ represents the Euclidean distance between bins $a$ and $b$. The parameters $\alpha$ and $\beta$ were randomly chosen from [-3, -0.75] and [1, 10], respectively. Next, we simulated duplications by designating contiguous ranges of bins as duplicated regions in each Hi-C matrix $E$, and summing up the interaction frequencies of duplicated bins in $E$ to obtain the collapsed matrix $C$ of size $N_c \times N_c$. To evaluate ec3D's ability to reconstruct structures where duplicated regions fold into different conformations, we finally designed our simulations in a way that half of the samples had the same local substructures for the duplicated regions, while the other half had different local substructures.

Using the procedure described above, we randomly generated 10 structures for each combination of $k \in \{1, 2, 3\}$ and $N_e \in \{250, 500, 750\}$, which led to 90 structures in total. And for each simulated structure, we generated 5 expanded matrices $E$ without duplication and 5 collapsed matrices $C$ with duplication by varying $\alpha$ and $\beta$, giving 450 expanded matrices and 450 collapsed matrices.

## Performance metrics (RMSD, PCC)

Similar to other 3D reconstruction methods, we measure the (dis) similarity between two 3D structures $X$ and $X\prime$ of the same size $N$ through root mean squared distance (RMSD) and Pearson correlation

coefficient (PCC):

$$RMSD(X, X') = \sqrt{\frac{1}{N} \sum_{i=1}^{N} (X_i - X'_i)^2},$$

$$PCC(X, X') = \frac{\sum_{i,j} (d_{ij} - \bar{d})(d'_{ij} - \bar{d}')}{\sqrt{\sum_{i,j} (d_{ij} - \bar{d})^2 \sum_{i,j} (d'_{ij} - \bar{d}')^2}},$$

where $d_{ij}$ is the off-diagonal ($i \neq j$) Euclidean distance between bin $i$ and bin $j$, and $\bar{d} = \frac{1}{N^2} \sum_{i,j} d_{ij}$.

However, due to the flexibility of coordinates with respect to rigid transformation in reconstruction, we first aligned $X$ and $X'$ by translation, scaling, and rigid-body rotation using the Kabsch-Umeyama algorithm[65]. For a brief summary, the algorithm works in three steps. (i) Move the centroid of both structures to the origin by subtracting $X$ and $X'$ with their centroid $\bar{X}$ and $\bar{X}'$. (ii) Rescale the two structures by their maximum diameters. (iii) Rotate $X$ by singular value decomposition (SVD) to align it with $X'$ in the optimal orientation. Namely, we computed SVD of $X \cdot X'^T = VSW^T$ and rotated $X$ by $(W \cdot V^T) \cdot X$.

### Similarity of 3D structure in duplicated regions

We used a permutation test to measure the significance of similarity between the local structures of duplicated regions $D_1$ and $D_2$ on the same ecDNA, or corresponding to the local 3D reconstruction of the same genomic interval but from different samples (e.g., ecDNA and HSR amplicon of *MYCN*). Specifically, to compare duplicated regions $D_1$ and $D_2$ on the same ecDNA, we randomly sampled 5,000 regions $S_i$ ($i = 1, 2, \cdots, 5000$) of the same size from the same molecule, and computed the fraction of times, a random pair $(D_1, S_i)$ had a smaller RMSD or larger PCC compared to the duplicated pair $(D_1, D_2)$ as the empirical p-value. To compare 3D reconstruction of the same genomic interval from different samples, we sampled $S_i$ from the larger genome.

### Comparison of ec3D against other reconstruction methods

To compare ec3D and other methods (MiniMDS, ShRec3D, and PAS-TIS), we input the collapsed matrix $C$ to these tools and allow them to reconstruct the structure from simulations with default parameters. For simulations without duplication, we directly compared the reconstructed structures from each method to the ground truth structures. However, since these methods treat duplicated segments as single instances when processing the input matrix, they output a structure of size $N_c \times 3$, with only one copy of each bin when duplication exists. We therefore compared their reconstructed structures to a partial ground truth structure consisting of all unique bins and only the first copy of each duplicated bin.

### OligoSTORM library design

The *MDM2* library was designed to cover a 2 Mb region (roughly 1 Mb of *MDM2* ecDNA amplicon plus flanking regions to identify the chromosomal locus), specifically chr12: 68,000,001-70,000,000 (hg38). Genome homology regions for each oligo were designed using PaintSHOP[66] using the default settings (no repeats, 200 max off-target score, 5 max K-mer count, and minimum probe value 0), followed by appending of unique barcodes for each 200 kb interval with OligoLego[38] (https://github.com/gnir/OligoLego).

### OligoSTORM library amplification

Oligopaint libraries (Supplementary Data 10) were ordered from Twist Bioscience, whereas the primer, bridge, and toehold sequences were ordered from Integrated DNA Technologies (IDT, Supplementary Data 11) with standard desalting. The readout probes carrying an Alexa647N fluor at each end (two on each readout probe in total) were ordered from IDT with HPLC purification. The probe libraries were amplified with PCR as previously described[67]. Briefly, libraries were reconstituted to 20 ng μL$^{-1}$ and amplified via PCR (Kapa Hi-Fi PCR kit, Fisher 50-196-5217), cleaned (Zymo D4033/D4029), and eluted with ultra-pure water. The dsDNA product was then in vitro transcribed into RNA at 37 °C for 16 h (NEB HiScribe, E2040S). The RNA was then reverse transcribed into cDNA using Maxima H Minus Reverse Transcriptase (Thermo Fisher, EP0752). The remaining RNA was digested using 0.5 M NaOH and 0.25 M EDTA to obtain ssDNA, which was then cleaned and concentrated to a concentration of 200 pmol μL$^{-1}$. Product size was validated using a 2% agarose gel (ThermoFisher A45204).

### Sample preparation for sequential OligoSTORM imaging

Adherent TR14 cells were lifted with TripLE (Thermo Fisher, 12605010), diluted to a concentration of $1 \times 10^6$ cells per mL, and 150 μL of cell suspension was added to Ibidi single channel μ-Slides I 0.2 (Ibidi, 80167) coated with Poly-L-Lysine. Slides were left at 37 °C and 5% CO2 overnight to allow cell attachment. The following day, cell media was removed, cells were washed once with PBS and then fixed in 4% paraformaldehyde (16% PFA diluted to 4% with PBS, Electron Microscopy Services, 15710) solution for 10 min at room temperature. The fixed cells were washed and either used immediately or stored in PBS at 4 °C for up to two weeks. On the day of use, samples were washed with PBS followed by permeabilization of the membrane using 0.5% Triton-X (Sigma Aldrich, 93443) for 10 min at room temperature. From here, all subsequent incubations were done at room temperature on a shaker unless otherwise specified. Samples were washed with 1x PBST (1x PBS + 0.1% (v/v) Tween-20) for 2 min, 0.1 N HCl for 5 min, twice with 2x SSCT (2x SSC + 0.1% (v/v) Tween-20) for 1 minute, and last with 2x SSCT + 50% Formamide (v/v) for 2 min. The sample was then incubated with fresh 2x SSCT + 50% Formamide for 20 min on a heat block placed in a 60 °C water bath. Channel was then aspirated completely, and liquid was quickly replaced with a hybridization solution containing primary library probes (50% Formamide + 2x SSCT + 10% Dextran Sulfate (w/v) + 400 ng μL$^{-1}$ RNase A, + 4 pmol μL$^{-1}$ library). The sample containing the primary hybridization solution was then denatured for 3 min on a heat block in an 80 °C water bath. The sample with primary hybridization was left in a humidity chamber overnight at 42 °C. The following day, the primary hybridization solution was removed, and the sample was washed with 2x SSCT, followed by 4 washes of 5 min each on a heat block in a 60 °C water bath using SSCT prewarmed to 60 °C. After the warm washes, the sample was washed twice with room temperature 2x SSCT for 2 min each time, and then once with 1x PBS. At this point, gold nano-urchins (Sigma, 797707) were sonicated for 10 min and then diluted 1:30 with 1x PBS. The diluted nano-urchins were added to the sample and centrifuged at 500 x g for 3 min. The sample was then placed on the microscope and connected to a microfluidics system to be used for all subsequent secondary hybridizations.

### Sequential OligoSTORM imaging

Sequential OligoSTORM imaging was done using a Bruker Vutara VXL equipped with an Olympus 60x silicone objective with NA of 1.3. Oligopaint oligos were hybridized with Alexa Fluor 647 as well as an Alexa Fluor 405 activator molecule for photo-activation and were illuminated using 638 and 405 nm lasers (Omicron), respectively, and a 698/70 emission filter (Semrock bandpass). Blinking fluorophores were detected using a Hamamatsu Orca Fusion-BT camera with a 10 ms exposure time. At each round of imaging, we cycled through the axial dimension (Z) four times, taking 1000 frames per Z-step, with an approximate total depth of 5 μm at 100 nm intervals. Each sequential step was hybridized and imaged one at a time using a microfluidics system (Fluigent Bruker Integrated Perfusion System). For each round, secondary hybridization solutions (2x SSCT + 0.3% Tween-20 (v/v) + 30% Formamide (v/v) + 500 nM bridge oligo + 500 nM Alexa647

readout probe + 500 nM universal readout probe + 500 nM Alexa405) containing bridge oligos complimentary to one of the 200 kb barcoding regions, as well as a fluorescent readout sequence, was then added to the sample and incubated at room temperature in the dark for 1 h. A universal readout probe, which was complementary to the oligos' forward primer and therefore binds to the entire library, was added in the first sequential step only to be used to take a reference image and setting of Z-stacks. From the second step of imaging onwards, this oligo was replaced with 500 nM toeholds of the previous step to remove the previous bridge sequence. Following secondary hybridizations, the sample was washed 4 times for 5 min each with a wash buffer (2x SSCT + 35% Formamide (v/v)). Image buffer (10% (w/v) glucose, 2x SSC, 50 mM Tris, 1% (v/v) β-mercaptoethanol, and 2% (v/v) of a GLOX stock solution consisting of 100 mg/mL glucose oxidase (Sigma G2133-250KU), 7.5 mg/mL catalase (Sigma C40-500MG), 30 mM Tris, and 30 mM NaCl) is then added to the sample for imaging. Signal was localized as described previously[38]. Briefly, the precision of each localization event was determined, and localizations with an axial precision of worse than 100 nm were filtered out. After filtering of localizations, drift correction was done using the center of mass between fiducial markers across time points. Finally, localizations were clustered using DBSCAN[68] (with a search radius of 150 nm and 50 minimum points) to remove noise before subsequent cluster analysis.

### Identification of ecDNAs from sequential OligoSTORM imaging

We started with the clusters of the OligoSTORM localizations of a cell containing multiple *MDM2* ecDNAs (chr12:68647578-69657744) in the cell line TR14. The *MDM2* ecDNA covers 6 out of the 10 200-kb intervals, which correspond to 6 distinct clusters in the image. We first computed the centroid of each cluster, and then constructed a graph where nodes represent these centroids; edges were added between nodes if their Euclidean distance was below a specified cutoff. To identify individual ecDNAs, we searched for cliques (i.e., fully connected subgraphs) of size 6 (6-cliques) in this graph. We optimized the distance cutoff by grid search (100-1000 nm, stride=100) to maximize the number of 6-cliques. With a small distance cutoff, nodes remain disconnected and form no cliques; with excessive distances, spurious connections between signals from different ecDNAs may create false positive cliques that share nodes with the true cliques. The optimal cutoff 700 gives the maximum number of 6-cliques. We further removed noise and verified 10 cliques by aligning them to a reference image (Supplementary Fig. 11a). By overlaying them against a reference image of MDM2, we identified 10 cliques that were highly likely to represent *MDM2* ecDNA in images (Fig. 2h and Supplementary Fig. 11b).

### Comparison of ec3D prediction against OligoSTORM imaged structure

We had used ec3D to reconstruct the structure of the *MDM2* ecDNA at 5-kb resolution. To validate the ec3D structure, we collected the 3D coordinates of 40 consecutive 5-kb bins within each 200-kb region, and computed their centroids to obtain 6 aggregated coordinates in 3D space at 200-kb genomic resolution. We computed the distances between each pair of centroids (Supplementary Fig. 11c, lower triangle). Next, we computed the average distance for each pair of centroids in the 10 candidate ecDNA structures (cliques) described above (Supplementary Fig. 11c, upper triangle). We calculated the Pearson correlations between pairwise distances from the ec3D-reconstructed 3D structure and the imaged structures.

### Minimum bounding box analysis

A minimum volume bounding box can be used to describe the overall 3D shape of an ecDNA structure. We implemented both the "rotating calipers" method[69] and PCA to compute the bounding box. The rotating calipers method takes $O(N_e^3)$ time and computes an exact solution; PCA takes $O(N_e)$ time for the $N_e \times 3$ structure matrix $X$ and gives a good practical solution, though without approximation guarantee[70]. In fact, we mainly focused on the ratio between the largest dimension and the smallest dimension of the bounding box, which can separate disk-like structures from spherical structures. Both methods suggested similar ratios (reported in Results) - even if the optimum bounding boxes computed by rotating calipers turn out smaller than PCA bounding boxes.

We additionally tested if the reconstructed ecDNA structures could be placed into a flatter bounding box (i.e., with smaller edge length ratios) and still generate the observed Hi-C interactions. Specifically, we reconstructed 3D structures of the ecDNA by optimizing the objective function described in Eq. (5) with the scaling parameter $\beta$ fixed, but with the maximum range of the first axis repeatedly halving from [-1, 1] to $[-\frac{1}{32}, \frac{1}{32}]$. The other two axes remain in the range [-1, 1]. By fixing $\beta$, we ensured that the structure was not shrinking proportionally in all axes in reconstruction. Decreasing the range of one axis would not impact the Poisson likelihood of a disk-like structure, as bins could still be placed on a plane orthogonal to that axis, preserving the pairwise spatial distances; while for spherical structures, the Poisson likelihood would become worse, due to additional constraints in the 3D space disrupting expected spatial distances suggested by Hi-C interactions.

### Cell lines

Human glioblastoma GBM39/HSR tumor spheroid was derived from patient tissue[5]. Neuroblastoma cell line TR14 was a gift from J. J. Molenaar (Princess Máxima Center for Pediatric Oncology, Utrecht, Netherlands)[14]; cell line IMR-5/75 from F. Westermann (German Cancer Research Center, Heidelberg, Germany); cell line CHP-212 was obtained from the American Type Culture Collection (ATCC, Manassas, VA)[12]. Cell line identity was verified by STR genotyping (Genetica DNA Laboratories, Burlington, NC and IDEXX BioResearch, Westbrook, ME). Tumor material for RCMB56 was previously obtained with consent from a TP53-mutant SHH subgroup medulloblastoma of an eight-year-old male patient who was diagnosed at Rady Children's Hospital San Diego, under the protocol Molecular Tumor Profiling Platform for Oncology Patients (IRB 190055)[17]. Cell lines D458, H2170 and MSTO211H were obtained from ATCC (catalog numbers CRL-3632, CRL-5928, and CRL-2081).

### Hi-C data preparation

We downloaded the raw Hi-C data of CHP-212 and IMR-5/75 from Helmsauer et al.[12], D458 and RCMB56 from Chapman et al.[17], TR14 from Hung et al.[14], MSTO211H from Xie et al.[71]. We downloaded high coverage GM12878 and IMR90 Hi-C (as control samples without ecDNA amplification) from 4D nucleome (https://data.4dnucleome.org/). The Hi-C library for H2170 was prepared using the Arima-HiC kit. Hi-C libraries for GBM39EC and GBM39HSR were prepared following a standard protocol to investigate chromatin interactions[24]. Cells were crosslinked with 1% formaldehyde for 10 min at room temperature. After nuclei permeabilization, DNA was digested with 100 units of MboI, end-labeled with biotinylated nucleotides, and proximity-ligated. Samples were sequenced using Illumina NovaSeq in 150 bp paired-end reads, with 3 replications for both GBM39EC and GBM39HSR. We combined these replications into a single matrix in our structural reconstruction.

We processed the raw Hi-C reads with HiC-Pro version 3.1.0[59]. This process included aligning the reads to the human reference genome (hg38), removing duplicate reads, assigning reads to restriction fragments, filtering for valid interactions, and generating binned contact matrices. For Arima Hi-C, we set the restriction enzyme to ^GATC and G^ANTC, and trimmed 5 bases from the 5' end of both read 1 and read 2 before alignment as per their user guide. Otherwise, we set the restriction enzyme to MboI/Dpnii, and did not trim the reads. We

generated contact matrices at resolutions ranging from 2 kb to 1 Mb, but focused on 5 kb resolution mostly in our analysis, allowing for a detailed description of chromatin interactions. The HiC-Pro output was converted into *cooler* format (*.cool or *.mcool)[72] required by ec3D. Note that ec3D also supports *.hic format as input compatible for visualization and analysis with Juicebox tools[73], and internally converts *.hic input to *.cool format using hic2cool (https://github.com/4dn-dcic/hic2cool).

### ecDNA genome reconstruction from WGS data

Ec3D requires an ecDNA sequence as input. For GBM39, GBM39HSR, CHP-212, IMR-5/75 (HSR), H2170, and TR14, we assembled the ecDNA genomes from Oxford Nanopore WGS by running CoRAL[20]. We ran CoRAL with a non-default command line argument '--min_bp_support 10.0' (i.e., with minimum coverage cutoff 10 times the diploid coverage for breakpoints) to eliminate redundant breakpoints that could result from non-ecDNA structural variations or heterogeneous ecDNA sequences. We extracted the cycle with the largest predicted copy number from CoRAL's output as the (primary) ecDNA sequence. For RCMB56 and MSTO211H, we ran AmpliconArchitect[18] with default parameters on paired-end short reads and again selected the cycle with the largest CN as its ecDNA sequence. The resulting ecDNA sequence of RCMB56 also agreed with optical genome mapping (OGM) contigs[17]. For D458, we reused the ecDNA sequence from Chapman et al.[17] computed by AmpliconReconstructor[74] from WGS and OGM contigs. Compared with AmpliconArchitect output consisting of multiple small cycles (as part of the ecDNA sequence), OGM provided a single consensus ecDNA sequence of D458 that was supported by all informative contigs[17].

### RNA-seq data processing

We downloaded the RNA-seq data of CHP-212 from Boeva et al.[75]. For RNA-seq analysis, we aligned the reads to hg38 and called gene fusion events in ecDNA intervals using STAR-Fusion[76] (v1.14.0) with its built-in annotation GRCh38_gencode_v44_CTAT_lib_Oct292023.plug-n-play. We quantified the expression coverage for each 5 kb bin by first counting the coverage of each 50-bp bin using deepTools[77] v3.5.6 and then averaging the coverage within 5 kb bins. We ran multiple sequence alignment of reads supporting the *LPIN1-MIR3681HG* fused transcript and the readthrough event between breakpoints 11,785,247 (3') and 12,622,735 (5') on chr2 using Clustal Omega[78] (v1.2.4).

### CTCF ChIP-seq data processing and TAD boundary calling

We downloaded the CTCF ChIP-seq data of CHP-212 from Helmsauer et al.[12]. We first trimmed adapters (Trimmomatic[79] v0.39) and aligned the reads to hg38 (Bowtie[80] v2.5.4) with default parameters. We then counted read coverage using deepTools[77] (v3.5.6) by counting reads in non-overlapping 50-bp bins. For CTCF peak calling, the coverages were aggregated to identify 5-kb bins, with average coverage >= 30 called as CTCF peaks. For comparative TAD analysis between ec3D reconstructions and Hi-C, we generated a distance matrix by calculating $d_{ij}^{\alpha}$ for all pairs $i, j \in \{1, 2, \cdots, 338\}$, where $d_{ij}$ is the Euclidean distance between bin $i$ and bin $j$ and $\alpha$ is the exponent output by ec3D. TAD boundaries were called with FAN-C[81] v0.9.28 using insulation scores at 10 kb window size, respectively from the Hi-C matrix and distance matrix. Each TAD boundary also corresponded to a 5-kb bin on the ecDNA. We considered a TAD boundary to match a CTCF peak if the peak was located within ±5 kb of the boundaries.

We applied a hypergeometric test to assess the significance of matching between TAD boundaries and CTCF peaks. Let the total number of bins on ecDNA be $N$; and the total number of bins corresponding to CTCF peaks be $K$. The P-value of observing at least $k$ matching peaks from $n$ TAD boundaries is computed as the tail distribution $1 - \sum_{x=1}^{k} \Pr(X = x)$, where $\Pr(X = x) = \frac{\binom{K}{x}\binom{N-K}{n-x}}{\binom{N}{n}}$.

### Analysis of A/B compartmentalization

We generated three observed/expected (O/E) interaction matrices from CHP-212 Hi-C, CHP-212 spatial distance matrix (ec3D generated 3D structure), and GM12878 Hi-C at 5-kb resolution with Knight-Ruiz (KR) normalization. All three matrices covered the same genomic regions corresponding to the CHP-212 ecDNA. From each O/E matrix, we computed a correlation matrix and extracted its first principal component (PC1) to identify A/B compartmentalization. PC1 signs for CHP-212 ecDNA were assigned based on correlation with gene expression levels, and signs for GM12878 were assigned based on the distribution of intra-region interactions. A positive PC1 sign indicated an A (open) compartment, and a negative PC1 sign indicated a B (closed) compartment.

### Statistics & reproducibility

All ec3D runs were performed without fixing random seeds. Comparisons between ec3D and alternative methods on simulated data, analyses of reconstructions with limited axis ranges, and comparison between overexpressed and normally expressed genes were performed using one-sided Wilcoxon rank-sum tests. Differences in cumulative distributions of significant interactions between cancer cell lines and control samples were assessed using two-sample Kolmogorov-Smirnov tests. Structural similarity between duplicated segments was evaluated using the permutation test described in Methods. Hypergeometric test was applied to assess the significance of overlap between TAD boundaries and CTCF peaks. Exact p-values are reported where applicable.

No data were excluded from the analyses. No statistical method was used to predetermine sample size, as our analyses were primarily computational validations rather than biological experiments. The experiments were not randomized as they involved computational analysis of existing datasets. The Investigators were not blinded to allocation during experiments and outcome assessment.

### Reporting summary

Further information on research design is available in the Nature Portfolio Reporting Summary linked to this article.

## Data availability

The newly generated Hi-C data for H2170, GBM39 and GBM39HSR have been deposited to NCBI and are available under SRA accession PRJNA1259562. The short-read WGS and Hi-C data for MSTO211H (from Xie et al.[71]), which were used in this study, are available on Figshare via https://doi.org/10.6084/m9.figshare.30629882. The previously published long read WGS data for GBM39 and GBM39HSR are available under SRA accession PRJNA1110283. The previously published long read WGS, Hi-C and CTCF ChIP-seq data for CHP-212 and IMR-5/75 are available under SRA accession PRJNA622577; RNA-seq data of CHP-212 are available under GEO accession GSE90683. The previously published short read WGS data for RCMB56 (PDX) and OGM contigs for D458 are available under SRA accession PRJNA1011359. The previously published Hi-C data for RCMB56 and D458 are available under GEO accession GSE240985. The previously published long read WGS for TR14 are available under SRA accession PRJNA670737; Hi-C data for TR14 are available under SRA accession PRJNA732417. The previously published G&T single-cell data for CHP212 are available on the European Genome Archive (EGA) under accession number EGAS50000000509. Source data are provided with this paper.

## Code availability

The source code of ec3D is publicly available on GitHub at https://github.com/AmpliconSuite/ec3D under BSD 3-Clause License. The specific version used in this manuscript (v1.0.0) has been archived at Zenodo with https://doi.org/10.5281/ZENODO.17082088[82].

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

## Acknowledgements

We thank Dr. Mahidhar Tatineni and Nicole Wolter from SDSC for providing detailed guidance on running ec3D on the SDSC expanse clusters. We thank Dr. Elias Rodriguez-Fos for preparing the single-cell RNA-seq data for CHP-212. We thank members from the Bafna, Chang, and Mischel laboratories as well as all members from team eDyNAmiC for helpful discussion and feedback. J.Y.S.L. is supported by the CPRIT Training Grant (RP210041). S.W. is supported by the Cancer Prevention and Research Institute of Texas (CPRIT, RR210034) and the American Cancer Society (CAT-24-1379043-01-CAT). G.N. is supported by the Cancer Prevention & Research Institute of Texas Grant ID RR210018 and UTMB startup Funds. A.G.H. is supported by the *Deutsche Krebshilfe* (German Cancer Aid) Mildred Scheel Professorship program 70114107. H.Y.C. is an Investigator of the Howard Hughes Medical Institute. This work was delivered as part of the eDyNAmiC team supported by the Cancer Grand Challenges partnership funded by Cancer Research UK (CGCATF-2021/100023 [S.W.]; CGCATF-2021/100012 [P.S.M. and H.Y.C.]; CGCATF-2021/100025 [V.B.]) and the National Cancer Institute (OT2CA278683 [S.W.]; OT2CA278688 [P.S.M. and H.Y.C.]; OT2CA278635 [V.B.]). V.B. was supported by NCI U24CA264379 and NIH R01GM114362.

## Author contributions

B.C., K.Z., C.L., and V.B. conceived the project and the initial ec3D algorithm. K.Z., C.L., and V.B. formulated the model and algorithm for solving structures with duplicated segments. K.Z. and C.L. developed the ec3D software. O.S.C. and L.C. provided Hi-C data of RCMB56 and D458 cell lines. K.K. and S.Z. provided Hi-C data of GBM39 and GBM39HSR cell lines. J.Y.S.L. performed cell synchronization for MSTO211H cell line. Y.X. and Y.J.K. performed Hi-C data generation and analysis for MSTO211H cell line. M.E.S. and A.G.H. provided the TR14 cell line, and J.A. and G.N. performed sequential OligoSTORM imaging of TR14 cell line. K.Z. and J.L. performed the sequence reconstruction of cell lines. B.C. prepared the Hi-C data. C.L. performed imaging analysis with input from K.Z. C.L. performed a simulation of ecDNA structures and Hi-C. B.C., K.Z., C.L. performed other analyses, including HSR reconstruction, significant interactions, bounding box analysis, functional analysis, and ensemble reconstructions. B.C., K.Z., C.L., and V.B.

wrote the manuscript with input from all authors. S.W., L.C., G.N., A.G.H., P.S.M., H.Y.C., and V.B. supervised this work.

## Competing interests

K.K. is an employee and stockholder of Amgen as of 09/15/2025. S.W. is a member of the scientific advisory board of Dimension Genomics Inc. P.S.M. is co-founder of Boundless Bio, Inc. He has equity and chairs the scientific advisory board, for which he is compensated. P.S.M. is also co-founder of S1 Oncology. He has equity, is on the board of directors, and consults for the company, for which he is compensated. V.B. is a co-founder, consultant, scientific advisory board member and has equity interest in Boundless Bio, Inc. and Abterra, Inc. The remaining authors declare no competing interests.

## Additional information

[1]Department of Computer Science and Engineering, University of California San Diego, La Jolla, CA, USA. [2]Department of Biochemistry and Molecular Biology, University of Texas Medical Branch, Galveston, TX, USA. [3]Department of Pediatric Hematology and Oncology, Charité-Universitätsmedizin Berlin, Berlin, Germany. [4]Department of Medicine, University of California San Diego, La Jolla, CA, USA. [5]Sanford Burnham Prebys Medical Discovery Institute, La Jolla, CA, USA. [6]Center for Personal Dynamic Regulomes, Stanford University, Stanford, CA, USA. [7]Howard Hughes Medical Institute, Stanford University, Stanford, CA, USA. [8]Department of Dermatology, Stanford University School of Medicine, Stanford, CA, USA. [9]Department of Pathology, Stanford University School of Medicine, Stanford, CA, USA. [10]Children's Medical Center Research Institute, University of Texas Southwestern Medical Center, Dallas, TX, USA. [11]Berlin Institute of Health, Berlin, Germany. [12]Experimental and Clinical Research Center, Max Delbrück Center for Molecular Medicine Charité-Universitätsmedizin Berlin, Berlin, Germany. [13]Sarafan Chemistry, Engineering, and Medicine for Human Health (Sarafan ChEM-H), Stanford University, Stanford, CA, USA. [14]Halıcıoğlu Data Science Institute, University of California San Diego, La Jolla, CA, USA. [15]Present address: New York Genome Center, New York, NY, USA. [16]Present address: Department of Neuro-oncology, Institute of Brain Sciences, Nagoya City University Graduate School of Medical Sciences, Nagoya, Japan. [17]Present address: Amgen Research, South San Francisco, CA, USA. [18]These authors contributed equally: Biswanath Chowdhury, Kaiyuan Zhu, Chaohui Li. ✉e-mail: vbafna@ucsd.edu

