## [Transparent Peer Review file · Nature Communications]

Reconstructing the three-dimensional architecture of extrachromosomal DNA with ec3D

Corresponding Author: Professor Vineet Bafna

Version 1:

Reviewer comments:

Reviewer #1

(Remarks to the Author)

Chowdhury et al. present ec3D, a computational tool to resolve/predict the 3D-structure of extrachromosomal DNA based on Hi-C data. The paper is well-written, and in principle addresses an interesting topic. However, there are significant limitations at the conceptual level, as well as in the manuscript text and the implementation of the code. Detailed comments regarding these limitations and potential improvements are outlined below.

General comments and concerns regarding the manuscript:

1. ec3D requires an already resolved ecDNA sequence to function properly, which is often not available, and limits the usefulness of the tool. This limitation could be made more clear and discussed in more detail.
2. There is also a lack of a discussion of the importance of 3D structure of ecDNA in the introduction section. Some ecDNAs are known to function mostly through "hubs", and through inter-regulatory interactions to promote oncogene activation, as mentioned briefly by the authors. Presumably, the 3D-organization of an individual ecDNA is less relevant in that context (?). The authors must provide more details explaining how generating 3D models can provide more insight into the functional consequences of ecDNAs.
3. A major limitation is the assumption that ecDNAs exist as a single, static structure. Hence, only a single, optimal structure is generated from a run of ec3D. This assumption is strong. Is there no way to estimate an ensemble of structures to provide some information about variability? The authors make rather strong claims about "oblate spheroidal" structures of ecDNA, as well as claiming that they engage in "crossing interactions" etc. But how is this interpreted considering that Hi-C data is generated in a population of cells?

(Remarks on code availability)

4. The authors claim that the tool supports files in Cooler and .hic format. However, this does not appear to be true. In the paper, the authors claim that "ec3D also supports *.hic format as input compatible for visualization and analysis with Juicebox tools, and internally converts *.hic input to *.cool format using hic2cool (<https://github.com/4dn-dcic/hic2cool>)". However, the only place where hic2cool is mentioned in the whole repository is in the readme, where the authors instruct users to convert their .hic files to .mcool using hic2cool. Thus, it is not accurate to state that "[ec3D] internally converts *.hic input to *.cool". Furthermore, converting a multi-resolution file with potentially several billion interactions just to fetch the small submatrices corresponding to ecDNAs seems rather inefficient. The authors could avoid this conversion altogether by using a format-agnostic library, such as hictkpy, to fetch interactions from .hic and .[m]cool files (see below):

```
diff --git a/setup.py b/setup.py
index 27816ee..5c25ef9 100644
--- a/setup.py
+++ b/setup.py
```

```

@@ -11,7 +11,7 @@ if __name__ == "__main__":
author_email = 'kaiyuan-zhu@ucsd.edu, chl221@ucsd.edu',
license = 'BSD 3-clause',
packages = find_packages('src'),
- install_requires = ['cooler',
+ install_requires = ['hictkpy',
'iced',
'scipy',
'scikit-learn',
diff --git a/src/extract_matrix.py b/src/extract_matrix.py
index 60aa512..126d109 100644
--- a/src/extract_matrix.py
+++ b/src/extract_matrix.py
@@ -4,7 +4,7 @@ Extract ecDNA matrix from whole genome Hi-C
import sys
import numpy as np
import argparse
-import cooler
+import hictkpy
import logging
import time
from iced import normalization
@@ -55,9 +55,9 @@ if __name__ == '__main__':
"""
Extract the expanded matrix from the input cool file
"""
- clr = cooler.Cooler(args.cool)
+ hf = hictkpy.File(args.cool, args.resolution)
chr_prefix = True
- if '1' in clr.chromnames:
+ if '1' in hf.chromosomes():
chr_prefix = False
row_labels = dict() # Map an interval of size RES to the index of all its copies
# Bins with negative orientation already reordered
@@ -97,7 +97,13 @@ if __name__ == '__main__':
intrvl_string_ = int2[0] + ":" + str(int2[1]) + "-" + str(int2[2])
if not chr_prefix:
intrvl_string_ = intrvl_string_[3:]
- mat_ = clr.matrix(balance = False, sparse = True).fetch(intrvl_string, intrvl_string_)
+
+ try:
+ mat_ = hf.fetch(intrvl_string, intrvl_string_, normalization=None).to_coo("full")
+ except RuntimeError as e:
+ if not str(e).endswith("overlaps with the lower-triangle of the matrix"):
+ raise
+ mat_ = hf.fetch(intrvl_string_, intrvl_string, normalization=None).to_coo("full").T
for i, j, v in zip(mat_.row, mat_.col, mat_.data):
D[s1 + i][s2 + j] = v
s2 += (int2[2] - int2[1]) // res

```

5. Which version of the code was used to produce the results presented in the paper?

The repository on GitHub does not currently have any releases (or tags).

I think the authors should tag a release before the paper is accepted.

Furthermore, I encourage the authors to take advantage of Zenodo's integration with GitHub to permanently archive releases and get a DOI for each release.

6. In the paper and in the README on GitHub the authors state that ecDNA intervals should be provided in (extended) BED format.

I am not aware of any "extended" BED format.

Furthermore, the file provided as an example on GitHub is not in BED format. Rather, the file is a plain TSV whose first three columns happen to represent 1D genomic coordinates.

I think the authors should either require a well-formed BED6+ file (where strand information is stored in the 6th column, and the 4th and 5th columns are not used), or not call the format BED at all, and state that the tool requires a 4-column TSV file with columns chrom, start, end, and strand in the file header.

7. In its current form, ec3D is not a proper python package: while the project has a setup.py file, that file is only used to define some metadata and dependencies, but none of the ec3D files are actually installed when running pip.

I recommend the author to make ec3D into a proper python package (by e.g. defining a pyproject.toml file in the project root, see here for more details) and ideally publish the package on PyPI and/or bioconda, as this would make the tool more

accessible for users.

8. Consider whether the --resolution CLI option is actually needed: if users are required to provide a single-resolution Cooler file anyway, the resolution information is already embedded in the .cool file itself.

9. As far as I can see, the main entrypoint for ec3D is the ec3D.py script located under the src/ folder on the GitHub repository. This file uses the subprocess to call other python scripts located in the same folder.

This approach is quite unusual for a tool written in pure python and presents several drawbacks that made it difficult to run the tool on my end:

a) Subprocesses are spawned in a way that is prone to leave zombie processes needlessly consuming resources if the program terminates abruptly. In my case, I restarted the tool a few times before I was happy with the input parameters. This resulted in my machine becoming unresponsive, which was due to hundreds of zombie processes using 100% of the CPU for no reason. I had to manually find and kill all those processes. If you absolutely need to use subprocesses, please make sure the code can handle abrupt interruptions (e.g. start processes with `sp.Popen` only inside with statements, or use one of the many wrappers, such as `sp.check_call()` that implement proper error handling.

b) In its current form, the code does not work with virtualenvs, which are a very common way to install bioinformatic tools.

Example:

```
python3 -m venv venv
venv/bin/pip install .
venv/bin/python src/ec3D.py ...
```

The last command fails almost immediately because the subprocesses are spawned using "python" as the interpreter (which in this case is the python interpreter installed system-wide, which lacks the dependencies installed using pip).

There are ways to query python for the path to the interpreter that is running the script.

I suggest that the authors restructure the package to follow a more standard layout and be more user-friendly:

- Use relative imports to import classes and functions (e.g. `from .expand_matrix import c_obj`)
- Use the multiprocessing module that comes with the Python standard library to take advantage of multiple CPU cores

10. I couldn't find a way to customize the number of CPU cores used by the tool, the tool seems to always use somewhere between 20 and 30 CPU cores. This should definitely be something the user has control over.

11. The code makes assumptions about the chromosome names and reference genome assembly.

For example, the `extract_matrix.py` assumes that the .cool file either has a chromosome named "1" or that chromosomes are prefixed with "chr".

If a user attempts to run the tool on files using a reference genome without Chromosome 1, the program will likely crash (because the code assumes that it is safe to discard the first three letters in the chromosome name, leading to out-of-bound errors or incorrect results).

Either the authors make it clear that the tool is only meant to be used e.g. on human/mouse Hi-C data where chromosomes are either named "1" or "chr1", or the code should be updated to accommodate chromosomes with arbitrary names (and it's the user responsibility to ensure that the Cooler and BED file refer to the same reference genome).

Reviewer #2

(Remarks to the Author)

The manuscript describes ec3D, a computational framework to reconstruct the 3D structure of ecDNA using Hi-C data. The method uses a Poisson likelihood model to infer spatial proximity between genomic bins, accounting for the circular topology and duplicated segments in ecDNA.

This is an interesting method and manuscript, which fills a gap in the current toolbox to analyse ecDNAs in tumours.

Specific comments

- The manuscript assumes a single dominant ecDNA molecule. While this is practical, it does not always reflect the heterogeneity observed in real tumours. The authors should discuss this limitation and its implications for interpreting results, where more than one unique ecDNA molecule is present.
- Negative binomial distribution - often used for count data distributions - is chosen for statistical testing of genomic interactions. Is this to model the 1D count data specifically or how does this relate to the modelled interaction frequency following a negative power-law distribution?
- p4: "The first 450 $N \times N$ matrices E are expanded matrices without duplicated bins". Isn't the expanded matrix with dup and collapsed without dups? Which one is Fig2b referring to?
- The method applies a regularization term to penalize uneven spacing between adjacent bins, but it is not clear to me what the biological interpretation and sensitivity are - this should be better justified or explored.
- While the simulation results are somewhat convincing, the manuscript would greatly benefit from more extensive benchmarking on real datasets with known 3D structures (e.g., from imaging).
- The inference of local folds is interesting and the mathematical reasoning is described in the methods. I could not find any direct comparison to experimentally observed local chromatin structures. How does it relate to e.g. CTCF-anchors, E:P interactions, AB compartment structures?

(Remarks on code availability)

The code is part of the amplicon architect toolset. While I have not tested the code explicitly, it appears to follow the same principles as the other ampliconarchitect tools.

Reviewer #3

(Remarks to the Author)

(Remarks on code availability)

Version 2:

Reviewer comments:

Reviewer #1

(Remarks to the Author)

In general, the authors did a great job addressing my concerns.

The new Figure 2i is nice, but some points are partially hidden/obscured in the top right corner. Additionally, the units on both axes are missing. Please include a P-value for the correlation, and clarify what "Pearson:" refers to — perhaps the Pearson correlation coefficient (?)

There is also a lack of legends on all axes in Figure 3a.

(Remarks on code availability)

My comment regarding efficiency concerns the conversion step (where hic2cool needs to read and then write all genome-wide interactions), not the fetching of interactions.

I've looked at the issue tracker on the hickpy repository and have not seen the GLIBC issue mentioned anywhere. If you have not already done so, you should reach out to the hickpy authors and let them know about the installation issues you are encountering.

Thanks for tagging a release and archiving the code on Zenodo.

Did the authors repeat all the analyses with the v1.0.0 version of the code?

v1.0.0 includes commits that have been authored after the initial paper submission.

If the author did not re-run the analyses with ec3D v1.0.0, then I think it is more appropriate to specify the commit hash of the code that was used to generate the results presented in the paper.

Regardless, I don't see v1.0.0, nor the DOI mentioned anywhere in the manuscript.

I believe this kind of information should be listed in the code availability section.

Reviewer #2

(Remarks to the Author)

All my comments have been answered satisfactorily by the authors.

(Remarks on code availability)

Point-to-Point Response for NCOMMS-25-14164A-Z

We sincerely thank the reviewers for their time and effort in reviewing our manuscript and providing valuable comments. We have addressed all the comments, which have greatly improved the manuscript. The following text presents detailed responses (with changes in the revised manuscript colored in blue) to all reviewers.

Reviewer #1 (Remarks to the Author):

Chowdhury et al. present ec3D, a computational tool to resolve/predict the 3D-structure of extrachromosomal DNA based on Hi-C data. The paper is well-written, and in principle addresses an interesting topic. However, there are significant limitations at the conceptual level, as well as in the manuscript text and the implementation of the code. Detailed comments regarding these limitations and potential improvements are outlined below.

General comments and concerns regarding the manuscript:

1. ec3D requires an already resolved ecDNA sequence to function properly, which is often not available, and limits the usefulness of the tool. This limitation could be made more clear and discussed in more detail.

Response. This is an excellent point, and we agree with the reviewer that ec3D requires a resolved sequence. We have designed it to work very well with other tools that we have developed, including AmpliconArchitect (AA) and CoRAL, which take whole genome sequencing (WGS) data and identify putative sequences. AA has been run on nearly 40000 whole genomes, and we have ecDNA structures for 3990 of them (by August 1, 2025, [AmpliconRepository.org](https://ampliconrepository.org)). Thus, ec3D can be run directly after running AA. The reviewer is correct that our method would require concurrent WGS data, however. We note that although Hi-C can be used to detect structural variations, reliable *ecDNA discovery/reconstruction* tools currently require WGS data, and we believe that users of our tool will have WGS available. In the future, we will consider developing methods that detect ecDNA and resolve its sequence and structure using Hi-C data.

We have updated the **Discussion** section to include this potential limitation of ec3D:

Ec3D requires an assembled ecDNA sequence to work with. We designed it to directly utilize the outputs of AmpliconArchitect or CoRAL, which use whole genome sequencing to automatically identify ecDNA and reconstruct plausible sequences. Thus, ec3D cannot be used without accompanying whole genome sequencing. Although Hi-C technology has been used to detect structural variation and 3D conformation, reliable ecDNA discovery currently requires WGS data. In future development, we will focus on methods that detect ecDNA and resolve its sequence and structure using Hi-C data.

2. There is also a lack of a discussion of the importance of 3D structure of ecDNA in the introduction section. Some ecDNAs are known to function mostly through "hubs", and through inter-regulatory interactions to promote oncogene activation, as mentioned briefly by the authors. Presumably, the 3D-organization of an individual ecDNA is less relevant in that context (?). The authors must provide more details explaining how generating 3D models can provide more insight into the functional consequences of ecDNAs.

Response. The reviewer raises a good point. We would like to argue that the 3D organization is actually quite important even when ecDNA functions through hub formation. While we tried to say it in the manuscript, we clearly did not get the point across. For example, outputting the 3D structure allows us to identify interactions that *cannot* be attributed to the structure. Therefore, the 3D structure can be used to identify potential sites of ecDNA-ecDNA interactions (**Supplementary Fig. 23-25**), reproduced below.

Here, the blue dots in panel (a) identify significant, genomically distant interactions that cannot be explained by structural variation and are likely due to conformational changes in ecDNA. Panel (b) identifies loci that are interacting due to proximity as predicted by ec3D. Ideally, the interactions in panels (a) and (b) must be identical, but Panel (b) is largely a subset of Panel (a). The differential interactions in panel (c) are ones that suggest spatial proximity, but the spatial proximity is not observed in the output structure. Differential interactions can appear either due to heterogeneity of structures or due to ecDNA-ecDNA interactions during hub formation. In this example, we indeed find a significant interaction (dashed box) that suggests an active enhancer region on one ecDNA interacting with *MYC* on another ecDNA. In Future collaborative work, we will work to validate these predictions.

To address the reviewer's comment further in the revision with a more clear example, we collaborated with Sihan Wu's lab to acquire Hi-C data for the ecDNA containing MSTO211H cell line in two specific conditions: 2 replicates in G1 phase, and 2 replicates in M phase of the cell cycle. We observed that the structure in the mitotic state is very different from the structure in the G1 phase of the cell cycle. It has been suggested that any hubs in the G phase are disrupted in the M phase¹. Thus, our result is proof of principle that 3D structures can be used to

check for changes in hub formation. In the revision, we have included the figure panel (**Figure 5a**) below.

Future research with our collaborators will aim to mechanistically elucidate this point. We have also used this experiment to provide data for Q3 (next question below), and have a new combined text in the revision that is provided in the answer to the next question.

To address the reviewer's point regarding the functional importance of ecDNA, we have added the following lines to the end of the **Introduction** section.

The 3D structures allowed for improved detection of CTCF binding, A/B compartmentalization, and enhancer-promoter interactions. They also pointed to putative sites of ecDNA-protein, ecDNA-ecDNA interactions, and ecDNA-chromosome tethering, thereby providing new avenues for exploring ecDNA biology.

3. A major limitation is the assumption that ecDNAs exist as a single, static structure. Hence, only a single, optimal structure is generated from a run of ec3D. This assumption is strong. Is there no way to estimate an ensemble of structures to provide some information about variability? The authors make rather strong claims about "oblate spheroidal" structures of

ecDNA, as well as claiming that they engage in "crossing interactions" etc. But how is this interpreted considering that Hi-C data is generated in a population of cells?

Response. The reviewer is correct that ecDNA structures are dynamic rather than static. Therefore, the most meaningful interpretation of ec3D output is the average (or *consensus*) structure. Consensus methods (including ShRec3d², MiniMDS³, and PASTIS⁴) are indeed popular, but there are also ensemble methods (including Chrom3D⁵, GEM⁶, and 3DMax⁷), which infer a population of structures. The two approaches have pros and cons as mentioned in the introduction of Varoquaux *et al.* 2023⁴, which provide a rationale for consensus methods. Specifically, consensus structures can be “...visually inspected and analyzed, and can be integrated in a straightforward manner with other data sources, such as gene expression, methylation, and histone modifications, which are also ensemble based.”(Varoquaux *et al.*).

Ensembles can be generated by sampling from the distribution of all possible structures. In this case, the output is best expressed in the form of statistics on the variability of positions. Alternatively, ensembles can be generated by starting from different, randomly initialized structures and providing a locally optimum structure for each random initialization. In this case, we are sampling from the space of locally optimal structures. We have updated the **Implementation Details** section in Methods to provide a description for the capability of ec3D to compute an ensemble of structures:

To avoid local minima due to non-convexity, we run the initialization and iterative optimization multiple (5 by default) times, with random initialization of X for running MDS (see below for the initialization of X). The user can choose to output the final X that leads to the best objective value as a consensus structure, or all final structures as an ensemble of structures.

We addressed the reviewer’s second concern regarding the interpretability of the consensus structures using ec3D’s ensemble generation capability and showed that consensus and ensembles together can indeed reveal biologically meaningful features. We collaborated with the Wu lab at UT Southwestern to acquire Hi-C data for the ecDNA containing MSTO211H cell line in two specific conditions: 2 replicates in G1 phase, and 2 replicates in M phase of the cell cycle. For each of these 4 datasets, we additionally generated an ensemble of 5 structures for a total of 20 possible structures. We found the following:

1. The strength of correlation in pairwise distances between computationally generated ensemble structures from any biological sample replicate is high.
2. The strength of correlation in pairwise distances between ensemble structures from two replicates from the same cell-cycle phase is high and generally matches the correlation of ensemble structures from the same replicate.
3. In contrast, the consensus (and ensemble) structures in the G-phase and the M-phase are very different, suggesting strongly that the structures can change based on external factors, including hub formation and ecDNA-protein interactions.

These results suggest that consensus structures are robust across replicates and supported by strong correlations between Hi-C contact frequencies and predicted distances. They form a

reasonable representation of the ensemble and can therefore be used to investigate other properties, such as the structures being oblate spheroidal, crossing interactions, and others. The mechanistic understanding of these findings for MSTO211H will be explored in future collaborative work.

In the revision, we have added a panel **Figure 5a** and **Supplementary Figure 18**, both shown below, and the following paragraphs.

The ensemble and consensus ecDNA structures change with biological context. Methods for DNA structure reconstruction generate either a consensus²⁻⁴ or an ensemble of structures⁵⁻⁷, with both methodologies being useful for different applications⁴. While ec3D was primarily designed to generate a consensus structure, it can be run with multiple random initializations to obtain an ensemble of structures, which is akin to sampling from a collection of locally optimal structures. To investigate this feature of ec3D, we acquired Hi-C data for the ecDNA-containing MSTO211H cell line in two specific conditions: 2 replicates in G1 phase, and 2 replicates in M phase. For each of these 4 datasets, we generated an ensemble of 5 structures for a total of 20 ec3D-derived structures. The structures revealed some interesting insights. First, the pairwise correlation, measured using PCC, between ensemble structures from any single biological replicate was high (**Supplementary Fig. 20**). Correspondingly, the pairwise correlation between ensemble structures from two different replicates from the same cell-cycle phase was also high and matched the correlation of ensemble structures within the same replicate in both G1 phase and M phase. By contrast, the consensus (and ensemble) structures from G1 phase and M phase were very different (**Fig. 5a**). The results suggest that the consensus structures are a good representation of the ensemble. Additionally, ecDNA structures are governed by external influences such as different ecDNA-protein interactions and ecDNA-chromosome tethering. Ec3D-derived consensus structures can reliably reveal those differences.

We have additionally edited the following paragraph in the **Discussion**.

The individual ecDNA molecules with the same sequence may additionally have slightly different structures. In the presence of this structural heterogeneity, we focused on providing one representative (i.e., *consensus*) structure. We observed a strong negative correlation between the number of Hi-C interactions in a pair of bins and their predicted spatial distance, across all molecules tested, providing confidence in the predicted consensus structure. The consensus structure can be used to generate hypotheses, design experiments, and for integration with other data types (e.g., ChIP-seq). Furthermore, consensus structures have proven useful in revealing the components of regulatory machinery in a region, including topologically associating domains (TADs) and chromatin loops⁸. While consensus methods are commonly used in 3D reconstruction, there are also ensemble methods for reconstruction, and ensemble structures provide complementary information. We briefly investigated ensembles by generating multiple locally optimal structures for the MSTO211H cell line. The results revealed that ensemble structures within the same biological condition (G1 or M) were highly similar to one another. In contrast, structures derived from G1 differed substantially from those in M phase. Thus, consensus structures provide a reliable representation of ensembles while also capturing relevant differences in different cell cycle phases. Additionally, ecDNA structures are governed by external influences such as different ecDNA-protein interactions and ecDNA-chromosome tethering. Future work will focus on the mechanistic underpinnings of these structural changes.

Reviewer #1 (Remarks on code availability):

4. The authors claim that the tool supports files in Cooler and .hic format. However, this does not appear to be true.

*In the paper, the authors claim that “ec3D also supports *.hic format as input compatible for visualization and analysis with Juicebox tools, and internally converts *.hic input to *.cool format using hic2cool (<https://github.com/4dn-dcic/hic2cool>)”.*

However, the only place where hic2cool is mentioned in the whole repository is in the readme, where the authors instruct users to convert their .hic files to .mcool using hic2cool.

*Thus, it is not accurate to state that “[ec3D] internally converts *.hic input to *.cool”.*

Response. Thanks for this comment. We have added a script to ec3D, which can automatically convert *.hic to *.cool if the input Hi-C data is in the former format.

Furthermore, converting a multi-resolution file with potentially several billion interactions just to fetch the small submatrices corresponding to ecDNAs seems rather inefficient. The authors could avoid this conversion altogether by using a format-agnostic library, such as hictkpy, to fetch interactions from .hic and .[m]cool files (see below):

```
diff --git a/setup.py b/setup.py
index 27816ee..5c25ef9 100644
--- a/setup.py
+++ b/setup.py
```

```

@@ -11,7 +11,7 @@ if __name__ == "__main__":
author_email = 'kaiyuan-zhu@ucsd.edu, chl221@ucsd.edu',
license = 'BSD 3-clause',
packages = find_packages('src'),
- install_requires = ['cooler',
+ install_requires = ['hictkpy',
'iced',
'scipy',
'scikit-learn',
diff --git a/src/extract_matrix.py b/src/extract_matrix.py
index 60aa512..126d109 100644
--- a/src/extract_matrix.py
+++ b/src/extract_matrix.py
@@ -4,7 +4,7 @@ Extract ecDNA matrix from whole genome Hi-C
import sys
import numpy as np
import argparse
- import cooler
+ import hictkpy
import logging
import time
from iced import normalization
@@ -55,9 +55,9 @@ if __name__ == '__main__':
"""
Extract the expanded matrix from the input cool file
"""
- clr = cooler.Cooler(args.cool)
+ hf = hictkpy.File(args.cool, args.resolution)
chr_prefix = True
- if '1' in clr.chromnames:
+ if '1' in hf.chromosomes():
chr_prefix = False
row_labels = dict() # Map an interval of size RES to the index of all its copies
# Bins with negative orientation already reordered
@@ -97,7 +97,13 @@ if __name__ == '__main__':
intrvl_string_ = int2[0] + ":" + str(int2[1]) + "-" + str(int2[2])
if not chr_prefix:
intrvl_string_ = intrvl_string_[3:]
- mat_ = clr.matrix(balance = False, sparse = True).fetch(intrvl_string, intrvl_string_)
+
+ try:
+ mat_ = hf.fetch(intrvl_string, intrvl_string_, normalization=None).to_coo("full")
+ except RuntimeError as e:
+ if not str(e).endswith("overlaps with the lower-triangle of the matrix"):

```

```
+ raise
+ mat_ = hf.fetch(intrvl_string_, intrvl_string, normalization=None).to_coo("full").T
for i, j, v in zip(mat_.row, mat_.col, mat_.data):
D[s1 + i][s2 + j] = v
s2 += (int2[2] - int2[1]) // res
```

Response. We thank the reviewer for their constructive suggestion. Hi-C data are usually sequenced and stored at a whole-genome scale, and extracting ecDNA submatrices from whole-genome Hi-C usually takes several seconds in ec3D, which is acceptable in terms of efficiency. We also tried giving users an option to use hictpky to fetch interactions from *.hic or *.m.cool file, but the installation of the latest version (v1.3.0) failed due to a GLIBC version compatibility issue raised by CMake. We successfully installed an old version (v0.0.5) of hictpky, but it failed to run following the reviewer's suggestion. We will add the hictpky library to ec3D when the issue is fixed.

5. Which version of the code was used to produce the results presented in the paper?

The repository on GitHub does not currently have any releases (or tags).

I think the authors should tag a release before the paper is accepted.

Furthermore, I encourage the authors to take advantage of Zenodo's integration with GitHub to permanently archive releases and get a DOI for each release.

Response. Thanks for raising the important question about version control. All results presented in the manuscript were produced by ec3D v1.0.0. We have added the release tag in the GitHub repository. We have also archived the release and generated a DOI via Zenodo. We will maintain the codebase, keep updating our program, and publish new releases on an ongoing basis.

6. In the paper and in the README on GitHub the authors state that ecDNA intervals should be provided in (extended) BED format. I am not aware of any "extended" BED format. Furthermore, the file provided as an example on GitHub is not in BED format. Rather, the file is a plain TSV whose first three columns happen to represent 1D genomic coordinates.

I think the authors should either require a well-formed BED6+ file (where strand information is stored in the 6th column, and the 4th and 5th columns are not used), or not call the format BED at all, and state that the tool requires a 4-column TSV file with columns chrom, start, end, and strand in the file header.

Response. Thanks for pointing out the inconsistency with the BED format. We have removed the word "extended" and rewritten the sentence in the overview as follows: The input ecDNA sequence is represented by ordered and oriented genomic segments in Browser Extensible Data (bed) format. We have revised the code to take the BED6+ (Genome Browser FAQ) format, where strand information is stored in the 6th column as input.

7. In its current form, ec3D is not a proper python package: while the project has a setup.py file, that file is only used to define some metadata and dependencies, but none of the ec3D files are

actually installed when running pip. I recommend the author to make ec3D into a proper python package (by e.g. defining a pyproject.toml file in the project root, see here for more details) and ideally publish the package on PyPI and/or bioconda, as this would make the tool more accessible for users.

Response. We thank the reviewer for the suggestion on usability. We have defined a pyproject.toml file and published the ec3D Python package on PyPI. Users can directly use ec3D after downloading ec3D with the command “pip install ec3D”. We will maintain the codebase and publish new releases on an ongoing basis.

8. Consider whether the --resolution CLI option is actually needed: if users are required to provide a single-resolution Cooler file anyway, the resolution information is already embedded in the .cool file itself.

Response. Thanks for pointing out this detail. If a *.hic or *.mcool file is provided, the user needs to specify the resolution since the Hi-C file may contain data in multiple resolutions. The resolution information is also required for the visualization, and it helps users ensure that all the computation is in the intended resolution. We also added input validation to check consistency between data resolution and user-specified resolution if a single-resolution Cooler is input.

9. As far as I can see, the main entrypoint for ec3D is the ec3D.py script located under the src/ folder on the GitHub repository. This file uses the subprocess to call other python scripts located in the same folder. This approach is quite unusual for a tool written in pure python and presents several drawbacks that made it difficult to run the tool on my end:

a) Subprocesses are spawned in a way that is prone to leave zombie processes needlessly consuming resources if the program terminates abruptly. In my case, I restarted the tool a few times before I was happy with the input parameters. This resulted in my machine becoming unresponsive, which was due to hundreds of zombie processes using 100% of the CPU for no reason. I had to manually find and kill all those processes. If you absolutely need to use subprocesses, please make sure the code can handle abrupt interruptions (e.g. start processes with sp.Popen only inside with statements, or use one of the many wrappers, such as sp.check_call() that implement proper error handling.

Response. We thank the reviewer for pointing out the issue caused by subprocesses. We have replaced all subprocesses with function calls in the main entry point of ec3D, preventing the creation of zombie processes. We also added user control on the maximum number of threads used by ec3D to prevent using up all CPU cores (see our response to question #10 below). We appreciate this comment for helping improve our software usability.

b) In its current form, the code does not work with virtualenvs, which are a very common way to install bioinformatic tools.

Example:

```
python3 -m venv venv  
venv/bin/pip install .
```

`venv/bin/python src/ec3D.py ...`

The last command fails almost immediately because the subprocesses are spawned using “python” as the interpreter (which in this case is the python interpreter installed system-wide, which lacks the dependencies installed using pip).

There are ways to query python for the path to the interpreter that is running the script.

I suggest that the authors restructure the package to follow a more standard layout and be more user-friendly:

- Use relative imports to import classes and functions (e.g. from `.expand_matrix` import `c_obj`)*
- Use the multiprocessing module that comes with the Python standard library to take advantage of multiple CPU cores*

Response. Thanks for raising the issue in Python virtual environments. As responded above, we have restructured the ec3D package and replaced subprocess spawning with relative imports. We also added user control over the maximum number of threads so that ec3D makes full use of available computational resources. Moreover, we included all dependencies of ec3D in `pyproject.toml` and uploaded an executable package to PyPI. As a result, users can install ec3D from PyPI by running “`pip install ec3D`” or install locally “`pip install .`” in a Python virtual environment, where ec3D can work well as suggested by the reviewer.

10. I couldn't find a way to customize the number of CPU cores used by the tool, the tool seems to always use somewhere between 20 and 30 CPU cores. This should definitely be something the user has control over.

Response. We thank the reviewer for raising the concern about CPU usage. The multi-core utilization arises from parallelized BLAS (Basic Linear Algebra Subprograms) operations within `scipy.optimize` (for reconstructing 3D structures), which may use up all available CPU cores for matrix operations. We have added an upper limit on the number of threads used by ec3D. Each thread created by BLAS operations typically uses one CPU core, so the thread number limit can control the number of CPU cores. A CLI option `-num_threads` (8 by default) has been added in ec3D for user customization. We have tested this approach and confirmed it successfully limits CPU core usage as expected.

11. The code makes assumptions about the chromosome names and reference genome assembly. For example, the `extract_matrix.py` assumes that the `.cool` file either has a chromosome named “1” or that chromosomes are prefixed with “chr”. If a user attempts to run the tool on files using a reference genome without Chromosome 1, the program will likely crash (because the code assumes that it is safe to discard the first three letters in the chromosome name, leading to out-of-bound errors or incorrect results). Either the authors make it clear that the tool is only meant to be used e.g. on human/mouse Hi-C data where chromosomes are either named “1” or “chr1”, or the code should be updated to accommodate chromosomes with arbitrary names (and it's the user responsibility to ensure that the Cooler and BED file refer to the same reference genome).

Response. Thank you for raising the issue about chromosome names. We have added exception handling to capture possible errors caused by invalid chromosome names and give users a prompt like “Chromosome names must be either chr1 or 1”. We have also specified in the README the convention of chromosome names used by ec3D.

Reviewer #2 (Remarks to the Author):

The manuscript describes ec3D, a computational framework to reconstruct the 3D structure of ecDNA using Hi-C data. The method uses a Poisson likelihood model to infer spatial proximity between genomic bins, accounting for the circular topology and duplicated segments in ecDNA. This is an interesting method and manuscript, which fills a gap in the current toolbox to analyse ecDNAs in tumours.

Response: We thank the reviewer for their supportive comment.

Specific comments

- The manuscript assumes a single dominant ecDNA molecule. While this is practical, it does not always reflect the heterogeneity observed in real tumours. The authors should discuss this limitation and its implications for interpreting results, where more than one unique ecDNA molecule is present.

Response. We thank the reviewer for this very important comment. There are two sources of heterogeneity: **sequence heterogeneity** and **structural heterogeneity**.

Sequence heterogeneity can be observed when the dominant cycle identified by sequence reconstruction tools, such as AmpliconArchitect (AA) or CoRAL, does not explain most of the copy number. This information is known to ec3D, which uses the dominant cycle to generate a 3D structure and identifies significant interactions that are not supported by the main structure but by structural variations. In the revision, we have added an ec3D analysis of an ecDNA in cell line TR14 (**Supplementary Figure 22**, included below) and additional text to say how ec3D can corroborate sequence heterogeneity.

Ec3D predicts interactions that corroborate heterogeneity of ecDNA sequence. Sequence heterogeneity exists when different copies of ecDNA do not have identical sequences. It is observed when the dominant cycle identified by sequence reconstruction tools, such as AmpliconArchitect (AA) or CoRAL, does not explain all of the copy number change⁹. This information is known to ec3D, which uses the dominant cycle to generate a 3D structure and identifies significant interactions. All ecDNA-positive cell lines except TR14 carried a single dominating ecDNA species. A CoRAL reconstruction of a TR14 ecDNA carrying *MYCN* suggested 3 overlapping cycles with estimated copy numbers 279 (explaining 65.7% of the amplification), 86 (20.2%), and 30 (**Supplementary Fig. 22a**). We analyzed ec3D predictions of these ecDNA cycles to investigate sequence heterogeneity. Ec3D-derived structure using Cycle 1 (**Supplementary Fig. 22b**) revealed many significant interactions. Some were mediated by 3D topological constrictions (**Supplementary Fig. 22b,c**). In contrast, the interactions marked SV1-3 were not supported by the ec3D conformation. However, they were proximal to breakpoints created by Cycle 2, thereby independently corroborating its existence. Cycle 3 has a much lower copy number that explains only 5.8% of the amplicon, causing the reduced strength of the SV4 interaction.

The question of structural heterogeneity is equally important, and in the revision, we did another experiment to discuss its implications. This question was also asked by Reviewer 1, who suggested ensembles of structures to describe heterogeneity. We have reproduced the response to Reviewer 1 below.

We addressed the reviewer's second concern regarding the interpretability of the consensus structures using ec3D's ensemble generation capability and showed that consensus and ensembles together can indeed reveal biologically meaningful features. We collaborated with the Wu lab at UT Southwestern to acquire Hi-C data for the ecDNA containing MSTO211H cell line in two specific conditions: 2 replicates in G1 phase, and 2 replicates in M phase of the cell cycle. For each of these 4 datasets, we additionally generated an ensemble of 5 structures for a total of 20 possible structures. We found the following:

1. The strength of correlation in pairwise distances between computationally generated ensemble structures from any biological sample replicate is high.
2. The strength of correlation in pairwise distances between ensemble structures from two replicates from the same cell-cycle phase is high and generally matches the correlation of ensemble structures from the same replicate.
3. In contrast, the consensus (and ensemble) structures in the G-phase and the M-phase are very different, suggesting strongly that the structures can change based on external factors, including hub formation and ecDNA-protein interactions.

These results suggest that consensus structures are robust across replicates and supported by strong correlations between Hi-C contact frequencies and predicted distances. They form a reasonable representation of the ensemble and can therefore be used to investigate other properties, such as the structures being oblate spheroidal, crossing interactions, and others. The mechanistic understanding of these findings for MSTO211H will be explored in future collaborative work.

In the revision, we have added a panel **Figure 5a** and **Supplementary Figure 18**, both shown below, and the following paragraphs.

The ensemble and consensus ecDNA structures change with biological context. Methods for DNA structure reconstruction generate either a consensus²⁻⁴ or an ensemble of structures⁵⁻⁷, with both methodologies being useful for different applications⁴. While ec3D was primarily designed to generate a consensus structure, it can be run with multiple random initializations to obtain an ensemble of structures, which is akin to sampling from a collection of locally optimal structures. To investigate this feature of ec3D, we acquired Hi-C data for the ecDNA-containing MSTO211H cell line in two specific conditions: 2 replicates in G1 phase, and 2 replicates in M phase. For each of these 4 datasets, we generated an ensemble of 5 structures for a total of 20 ec3D-derived structures. The structures revealed some interesting insights. First, the pairwise correlation, measured using PCC, between ensemble structures from any single biological replicate was high (**Supplementary Fig. 20**). Correspondingly, the pairwise correlation between ensemble structures from two different replicates from the same cell-cycle phase was also high and matched the correlation of ensemble structures within the same replicate in both G1 phase and M phase. By contrast, the consensus (and ensemble) structures from G1 phase and M phase were very different (**Fig. 5a**). The results suggest that the consensus structures are a good representation of the ensemble. Additionally, ecDNA structures are governed by external influences such as different ecDNA-protein interactions and ecDNA-chromosome tethering. Ec3D-derived consensus structures can reliably reveal those differences.

We have additionally edited the following paragraphs in the **Discussion**.

Ec3D requires an assembled ecDNA sequence to work with. We designed it to directly utilize the outputs of AmpliconArchitect or CoRAL, which use whole genome sequencing to automatically identify ecDNA and reconstruct plausible sequences. Thus, ec3D cannot be used without accompanying whole genome sequencing. Although Hi-C technology has been used to detect structural variation and 3D conformation, reliable ecDNA discovery currently requires WGS data. In future development, we will focus on methods that detect ecDNA and resolve its sequence and structure using Hi-C data.

When counterpart WGS is available, tools like AmpliconArchitect and CoRAL often identify one dominant cycle that explains most of the observed copy number and structural variation. In this case, the predicted sequence (generated by traversing the cycle) is likely to be correct, and ec3D would work directly. In many of our analyzed samples, sequence heterogeneity was indeed low, and a single sequence explained the entire copy number. In other cases, multiple ecDNA species with varying sequences may exist. In these cases, ec3D provides the structure of a plausible ecDNA using the dominant cycle, but may also reveal evidence of sequence heterogeneity. In TR14, ec3D revealed sequence heterogeneity by identifying significant interactions near structural variants that were not part of the dominant cycle. Moreover, the strength of the interactions correlated with the copy numbers of the alternative ecDNA sequences. Ec3D can also provide information about the quality of ecDNA reconstruction by analysis of correlations between the ec3D predicted pairwise distances and the Hi-C generated contact frequencies. Recently, Zhao *et al.*¹⁰ have utilized ec3D to support the existence of two different ecDNA species in a Glioblastoma tumor sample. In future work, we also plan to develop an integrated tool for identifying the ecDNA sequence using both WGS and Hi-C data. When multiple distinct ecDNA species exist, novel methods will be required to deconvolute the Hi-C data to provide a consensus structure for each species.

The individual ecDNA molecules with the same sequence may additionally have slightly different structures. In the presence of this structural heterogeneity, we focused on providing one representative (i.e., *consensus*) structure. We observed a strong negative correlation between the number of Hi-C interactions in a pair of bins and their predicted spatial distance, across all molecules tested, providing confidence in the predicted consensus structure. The consensus structure can be used to generate hypotheses, design experiments, and for integration with other data types (e.g., ChIP-seq). Furthermore, consensus structures have proven useful in revealing the components of regulatory machinery in a region, including topologically associating domains (TADs) and chromatin loops⁸. While consensus methods are commonly used in 3D reconstruction, there are also ensemble methods for reconstruction, and ensemble structures provide complementary information. We briefly investigated ensembles by generating multiple locally optimal structures for the MSTO211H cell line. The results revealed that ensemble structures within the same biological condition (G1 or M) were highly similar to one another. In contrast, structures derived from G1 differed substantially from those in M phase. Thus, consensus structures provide a reliable representation of ensembles while also capturing relevant differences in different cell cycle phases. Additionally, ecDNA structures are governed

by external influences such as different ecDNA-protein interactions and ecDNA-chromosome tethering. Future work will focus on the mechanistic underpinnings of these structural changes.

- *Negative binomial distribution - often used for count data distributions - is chosen for statistical testing of genomic interactions. Is this to model the 1D count data specifically or how does this relate to the modelled interaction frequency following a negative power-law distribution?*

Response. We thank the reviewer for raising the question about the difference between the statistical models we used to identify significant interactions and to reconstruct 3D structures. We do not explicitly investigate the relationship between these two statistical frameworks, as they serve different purposes in ec3D's reconstruction pipeline.

- Negative binomial distribution is used specifically to model the “background” interaction frequency for *all bin pairs with fixed genomic distance*. And the goal is to identify statistically significant interactions (relative to this background model) from the Hi-C count data.
- Negative power-law distribution is used separately to model observed interactions as a function of spatial distance *for each bin pair*, which enables 3D reconstruction from Hi-C.

We have modified the Methods section to clarify that the negative binomial distribution is only used to identify significant interactions:

Specifically, to identify statistically significant interactions, we always model the interaction frequencies at each genomic distance g using a negative binomial distribution with mean μ_g and variance σ_g ($\sigma_g > \mu_g$).

- p4: *"The first 450 $N \times N$ matrices E are expanded matrices without duplicated bins". Isn't the expanded matrix with dup and collapsed without dups? Which one is Fig2b referring to?*

Response. We thank the reviewer for pointing out this confusing sentence. In an expanded matrix, duplicated bins are represented as multiple separate entries, resulting in larger $N_e \times N_e$ matrices. Collapsed matrices, on the other hand, aggregate signals from duplicated bins by summing up their interaction counts, resulting in smaller $N_c \times N_c$ matrices where each unique bin occurs only once. We have corrected these two sentences as follows:

The first 450 $N_e \times N_e$ matrices E are expanded matrices with duplicated bins kept separate. The other 450 $N_c \times N_c$ matrices C are collapsed matrices after merging duplicated bins.

We have also removed the irrelevant pointer to Fig. 2b in the above sentence, which refers generally to a simulated Hi-C matrix, and indicates that ec3D actually takes this matrix to reconstruct the structure.

- The method applies a regularization term to penalize uneven spacing between adjacent bins, but it is not clear to me what the biological interpretation and sensitivity are - this should be better justified or explored.

Response. We thank the reviewer for this important question, which is not well discussed in our current manuscript. The goal of regularization is to penalize large spatial discontinuities between adjacent bin pairs. The biological rationale is that chromatin exists as a continuous linear polymer chain. As connected beads in a polymer, two neighboring genomic loci should have roughly equal spatial distance in the 3D space. This principle is particularly useful for separating interaction signals from duplicated region(s). Duplicated genomic segments on the same ecDNA are assumed to occupy different spatial locations. The spacing constraint hence helps distinguish between these copies by ensuring that each maintains local polymer continuity, preventing potential erroneous “jumping” between spatially distant copies of the same sequence. We have added the following text to the revision.

In our model, ecDNA chromatin forms a continuous polymer chain, where fixed resolution bins are represented as discrete points along the chain. To ensure two adjacent bins have roughly equal spatial distance in the 3D space, we included a regularizer term (see **Methods** and **Supplementary Fig. 9**). The impact of this additional constraint on reconstruction was modest. The average absolute difference between the RMSD (and PCC) values with and without regularization was 0.0324 (and 0.0170) in the simulated data.

We also improved the corresponding description of methods to clarify the regularization.

“...since ecDNA forms a continuous polymer chain with fixed resolution bins as discrete points along the chain, we introduce a regularization term to control the variance between consecutive bins a and $a + 1$.”

- While the simulation results are somewhat convincing, the manuscript would greatly benefit from more extensive benchmarking on real datasets with known 3D structures (e.g., from imaging).

Response. This is a great comment, and one that we have been working on since our original submission. Imaging of ecDNA is not a solved problem, and major considerations must be resolved before a comparison can be made.

- There are multiple ecDNAs per cell, with tremendous count differences from cell to cell because of the independent segregation of ecDNA during mitosis. Thus, each cell is unique in terms of imaging.
- EcDNAs are relatively small and compact because of circularity, and the images may be diffraction-limited, with additional confounding due to overlapping ecDNA in the same region.
- Duplicated segments will also be challenging for imaging.
- The reconstruction of the 3-dimensional structure is preceded by the computation of distances between pairs of corresponding probes, and pairwise distances from different copies of the ecDNA must be integrated or averaged.

Despite these challenges, we wanted to address the reviewer’s concerns. Therefore, we initiated a collaboration with members of the Henssen lab who generously donated a cell line TR14, and with Prof. Guy Nir and Jessica Alsing (UT Medical Branch), who acquired a super-resolution image, with probes at 200kb intervals. Even this experiment required the generation of customized probes, two continuous weeks of image acquisition, and the development of methods to isolate imaging data corresponding to 10 MDM2 ecDNAs. The bottom line is that we obtained a good correlation between bin-pair distances predicted by ec3D and those estimated from 10 ecDNAs in the Imaging data (Pearson’s $R = 0.84$). To our knowledge, this is the first imaging analysis of an entire ecDNA. In the revised manuscript, we have added methods for (a) imaging data probe design and raw data acquisition; (b) imaging data initial centroiding; (c) imaging data ecDNA specific analysis, and (d) correlation of ec3D with imaging data. The distance correlations have been added to revised **Figure 2h, 2i**, and **Supplementary Figure 11** as shown below. We also moved the current **Figure 2h, 2i** panels to the supplementary.

h

i

The revised manuscript paragraph is included below.

Ec3D predicted structures correlate strongly with imaged ecDNAs. High-resolution microscopy using fluorescent probes has also been used to understand the 3-dimensional structure of DNA molecules^{11–13}, but has not been attempted previously for an entire ecDNA. Notably, the process is not trivial because each cell contains an unknown number of ecDNA molecules. To provide a baseline comparison, we chose the TR14 cell line containing the oncogene *MDM2*. We acquired Hi-C data and reconstructed the 3D structure using ec3D (see next section). Next, we designed custom probes covering the ecDNA region with 200-kb genomic resolution. The probes were imaged using Sequential OligoSTORM^{11,14} (see **Methods**). The centroids of fluorescent signals from each probe were used to identify putative ecDNAs in a cell (**Fig. 2h** and **Supplementary Fig. 11a-c**). Pairwise distances predicted by the OligoSTORM images strongly correlated with the corresponding pairwise distances predicted by ec3D (Pearson Correlation coefficient 0.84, **Fig. 2i**), providing a baseline, orthogonal validation of ec3D structures.

a

b

c

The methodology for probe design and image acquisition is described in **Methods** and the **Supplementary Figure 11**, shown above. The reference image showing the positions of the *MDM2* ecDNA is shown in panel (a). **Figure 2h** displays the raw data points corresponding to the 6 probes that tile the ecDNA. Panel (b) displays centroid cliques of size 6 that represent the *MDM2* ecDNA structures for each ecDNA. Panel (c) shows the pairwise probe distances that were averaged for comparison with the Hi-C derived structure.

We agree that more work will be needed to provide accurate comparisons of the two technologies, but in some sense, the tremendous effort and novel methods needed for the generation of ecDNA imaging data also justify the need for Hi-C based reconstruction as a relatively more accessible method. In the **Discussion**, we have revised the relevant paragraph as below:

The challenge of 3D reconstruction of DNA structures can potentially be addressed by other complementary methodologies. Multiplexed imaging of hundreds of genomic loci by sequential hybridization has the potential to elucidate 3-dimensional structures of entire chromosomes at single-cell levels, albeit typically with a tradeoff between throughput and genomic resolution^{12,13,15}. We therefore focused on chromatin capture data due to finer genomic resolution and ease of data acquisition, although the data was acquired from bulk and not single cells. We also acquired super-resolution imaging data at 200-kb genomic resolution to compare against ec3D structures. The results, averaged over 10 ecDNAs, revealed a strong correlation in the pairwise distances predicted by imaging and ec3D. In future experiments, we plan to further utilize multiplexed imaging with finer genomic resolution¹¹⁻¹³, to better understand ecDNA structure at the single-cell level.

- The inference of local folds is interesting and the mathematical reasoning is described in the methods. I could not find any direct comparison to experimentally observed local chromatin structures. How does it relate to e.g. CTCF-anchors, E:P interactions, AB compartment structures?

Response. We thank the reviewer for this valuable comment. Note that because ecDNA is small and distinct from chromosomes, the discussion of CTCF, A/B, and E-P is also local to a single ecDNA, and not genome-wide as is usual for other Hi-C analyses.

In the manuscript, we identified CTCF anchors on the CHP-212 ecDNA by calling peaks from CHIP-seq data of the CHP-212 cell line. We also predicted TAD boundaries using the Hi-C data and spatial distances from the ecDNA 3D structure, respectively (**Figure 4a**, reproduced below). As TAD structures are often supported by CTCF binding sites, we measured the proximity of TAD boundaries to CTCF anchors. Ec3D-predicted TAD boundaries explained 53.85% of the top CTCF peaks (P-value: 4.5e-10) in comparison to the 26.92% explained using Hi-C (P-value:0.0029).

In the revision, we have further addressed the reviewer's question by investigating other structural and functional properties of the ec3D reconstructed structures, including A/B compartmentalization on the cell line CHP-212 identified from original Hi-C and spatial distances from the 3D structure. The panel **Figure 4b** and text have been added.

We next compared the A/B compartments of the CHP-212 ecDNA that were generated using either the original Hi-C or the spatial distances from the 3D structure (**Methods**). Compared to Hi-C, ec3D generated a correlation matrix with more intense signals. Further, ec3D identified similar but finer A/B compartment structures on CHP-212 ecDNA (**Fig. 4b, Supplementary Fig. 17**). We also investigated the A/B compartments of the identical genomic regions on a control cell line, GM12878. In direct contrast with GM12878, three distinct regions spanning chr2:15.58-16.07Mb, chr2:12.46-12.62Mb, and chr2:12.62-12.75Mb were in the same compartment on the CHP-212 ecDNA. We next explored the 3D structure of this compartment as predicted by ec3D (**Fig. 4c**) and the activity of its constituent genes.

We address the reviewer's comment regarding E-P interactions by showing how changes in gene expression and hijacking of super-enhancers were clarified by ec3D structure reconstruction. The text reads as below, and includes a novel finding of the split expression of *LPIN1* in the cell line CHP-212.

"Single-cell RNA-seq data of CHP-212 had previously revealed that out of the 6 genes present on ecDNA, 4 were overexpressed (*LPIN1*, *TRIB2*, *DDX1*, and *MYCN*), but the expression of two genes *GREB1*, *NTSR2* remained at basal level, with median expression at least 5X lower than the minimum median expression of the overexpressed genes (**Fig. 4d**)¹⁶. The ec3D structure clarifies this observation by revealing multiple topological domains on the A compartment

described above. One domain contained *LPIN1*, *TRIB2*, *DDX1*, and the other contained *MYCN* along with a super-enhancer, and this active compartment excluded the two basally expressed genes. Intriguingly, *LPIN1* was split in the ecDNA. The 5' half of *LPIN1* was over-expressed along with *TRIB2* and *DDX1*, while *LPIN1* 3' half was excluded and not-expressed (**Supplementary Fig. 18a**). Moreover, the structural variation that split *LPIN1*, brought the *LPIN1* 5' end in proximity to the 3' end of *MIR3681HG*, resulting in expression of a fused transcript (**Supplementary Fig. 18b**), and also a readthrough event resulting in expression of intronic DNA downstream of the genomic breakpoint (**Supplementary Fig. 18c**). Finally, the circularization removed the region immediately upstream of *GREB1*, consistent with its reduced expression. Thus, ecDNA can alter the regulation of genes through a combination of structural variation and 3D conformational change.

We also investigated a TAD on the Medulloblastoma cell line D458, which is a 2.5 Mb molecule amplifying the oncogenes *MYC* (chr8) and *OTX2* (chr14) on an ecDNA. Earlier results had suggested that a DNase-hypersensitive region (DHS1¹⁷) containing a putative enhancer located 80 kb from the *OTX2* gene on chr14 was essential for proliferation of the cell line¹⁸, and it was speculated that DHS1 might be hijacked by *MYC* to drive proliferation. However, DHS1 was found not to influence *MYC* activity on D458¹⁸; instead, it enhanced *OTX2* expression in other Medulloblastoma cell lines¹⁷. Ec3D analysis suggests a neo-TAD that includes DHS1, *OTX2*, and the lncRNA *OTX2-AS1*, but not *MYC*, providing more clarity for the observed experimental data (**Supplementary Fig. 19**). We also noted that an inversion of the *OTX2* region brought *OTX2-AS1* closer to the enhancer on the ecDNA, in contrast to their positioning on the reference genome.”

Reviewer #2 (Remarks on code availability):

The code is part of the amplicon architect toolset. While I have not tested the code explicitly, it appears to follow the same principles as the other ampliconarchitect tools.

Response. Reviewer 1 had raised important concerns and suggestions regarding the code, and we have taken those suggestions to improve it. As Reviewer 2 suggests, we are committed to supporting the codebase for ec3D on an ongoing basis and addressing any future issues as we have tried to do for AmpliconArchitect.

Reviewer #3 (Remarks to the Author):

Response. We thank the reviewer for their important service.

References

1. Hung, K. L. *et al.* ecDNA hubs drive cooperative intermolecular oncogene expression. *Nature* **600**, 731–736 (2021).
2. Lesne, A., Riposo, J., Roger, P., Cournac, A. & Mozziconacci, J. 3D genome reconstruction from chromosomal contacts. *Nat. Methods* **11**, 1141–1143 (2014).
3. Rieber, L. & Mahony, S. miniMDS: 3D structural inference from high-resolution Hi-C data. *Bioinformatics* **33**, i261–i266 (2017).
4. Varoquaux, N., Noble, W. S. & Vert, J.-P. Inference of 3D genome architecture by modeling overdispersion of Hi-C data. *Bioinformatics* **39**, btac838 (2023).
5. Paulsen, J. *et al.* Chrom3D: three-dimensional genome modeling from Hi-C and nuclear lamin-genome contacts. *Genome Biol.* **18**, 21 (2017).
6. Zhu, G. *et al.* Reconstructing spatial organizations of chromosomes through manifold learning. *Nucleic Acids Res.* **46**, e50–e50 (2018).
7. Oluwadare, O., Zhang, Y. & Cheng, J. A maximum likelihood algorithm for reconstructing 3D structures of human chromosomes from chromosomal contact data. *BMC Genomics* **19**, 161 (2018).
8. Dixon, J. R. *et al.* Topological Domains in Mammalian Genomes Identified by Analysis of Chromatin Interactions. *Nature* **485**, 376–380 (2012).
9. Zhu, K. *et al.* CoRAL accurately resolves extrachromosomal DNA genome structures with long-read sequencing. Preprint at <https://doi.org/10.1101/2024.02.15.580594> (2024).
10. Zhao, B. *et al.* Oncogenic drivers shape the tumor microenvironment in human gliomas. Preprint at <https://doi.org/10.1101/2025.05.16.654515> (2025).
11. Nir, G. *et al.* Walking along chromosomes with super-resolution imaging, contact maps, and integrative modeling. *PLOS Genet.* **14**, e1007872 (2018).
12. Su, J.-H., Zheng, P., Kinrot, S. S., Bintu, B. & Zhuang, X. Genome-Scale Imaging of the

- 3D Organization and Transcriptional Activity of Chromatin. *Cell* **182**, 1641-1659.e26 (2020).
13. Mateo, L. J. *et al.* Visualizing DNA folding and RNA in embryos at single-cell resolution. *Nature* **568**, 49–54 (2019).
 14. Bintu, B. *et al.* Super-resolution chromatin tracing reveals domains and cooperative interactions in single cells. *Science* **362**, eaau1783 (2018).
 15. Jia, B. B., Jussila, A., Kern, C., Zhu, Q. & Ren, B. A spatial genome aligner for resolving chromatin architectures from multiplexed DNA FISH. *Nat. Biotechnol.* **41**, 1004–1017 (2023).
 16. Stöber, M. C. *et al.* Intercellular extrachromosomal DNA copy number heterogeneity drives cancer cell state diversity. 2023.01.21.525014 Preprint at <https://doi.org/10.1101/2023.01.21.525014> (2024).
 17. Wortham, M. *et al.* Chromatin Accessibility Mapping Identifies Mediators of Basal Transcription and Retinoid-Induced Repression of OTX2 in Medulloblastoma. *PLoS ONE* **9**, e107156 (2014).
 18. Chapman, O. S. *et al.* Circular extrachromosomal DNA promotes tumor heterogeneity in high-risk medulloblastoma. *Nat. Genet.* **55**, 2189–2199 (2023).

Reviewer #1 (Remarks to the Author):

In general, the authors did a great job addressing my concerns.

Response: We thank the reviewer for their assessment of our revisions.

The new Figure 2i is nice, but some points are partially hidden/obscured in the top right corner. Additionally, the units on both axes are missing. Please include a P-value for the correlation, and clarify what “Pearson:” refers to — perhaps the Pearson correlation coefficient (?)

Response: We thank the reviewer for the comments. We have enlarged the axis ranges to show the points in the top right corner. We have clarified the P-value, Pearson correlation coefficient, and units of axes in the figure and legends.

The revised legends of **Figure 2i**: The correlation of the pairwise bin distances obtained by ec3D and by OligoSTORM imaging averaged across 10 ecDNA molecules. The distances were normalized to a range of 0.2 - 1.0 by min-max normalization. The P-value was calculated using a two-sided t-test.

There is also a lack of legends on all axes in Figure 3a.

Response: We have updated the legend of Figure 3a to describe the axes.

The revised legends of **Figure 3a**: Correlation between normalized Hi-C interaction frequencies (x-axis) and spatial (Euclidean) distances from ec3D reconstruction (y-axis) from GBM39. Each point in the plot corresponds to a pair of 5kbp bins. Color gradient representing the genomic distance was overlaid on each scatter plot point, with warmer colors indicating shorter genomic distances and cooler colors indicating longer distances. The Spearman and Pearson correlation

coefficients suggested a negative power law decay of interaction frequencies with spatial distances.

Reviewer #1 (Remarks on code availability):

My comment regarding efficiency concerns the conversion step (where hic2cool needs to read and then write all genome-wide interactions), not the fetching of interactions.

I've looked at the issue tracker on the hickpy repository and have not seen the GLIBC issue mentioned anywhere. If you have not already done so, you should reach out to the hickpy authors and let them know about the installation issues you are encountering.

Response: We thank the reviewer for the suggestion. We have fixed the installation issue due to CMake version inconsistency and will incorporate the tool in the next public version of ec3D.

Thanks for tagging a release and archiving the code on Zenodo.

Did the authors repeat all the analyses with the v1.0.0 version of the code?

v1.0.0 includes commits that have been authored after the initial paper submission.

If the author did not re-run the analyses with ec3D v1.0.0, then I think it is more appropriate to specify the commit hash of the code that was used to generate the results presented in the paper.

Regardless, I don't see v1.0.0, nor the DOI mentioned anywhere in the manuscript.

I believe this kind of information should be listed in the code availability section.

Response: We ran all the analyses with the v1.0.0 version before the code was published and tagged online. We have added the DOI and version number in the Code Availability section.

Reviewer #2 (Remarks to the Author):

All my comments have been answered satisfactorily by the authors.

Response: We thank the reviewer for their assessment of our revisions.